# A Review of Image Reconstruction Algorithms for Diffuse Optical Tomography

Shinpei Okawa * and Yoko Hoshi

Preeminent Medical Photonics Education & Research Center, Hamamatsu University School of Medicine, 1-20-1 Handayama, Higashi-ku, Hamamatsu 431-3192, Shizuoka, Japan
* Correspondence: okawa@hama-med.ac.jp; Tel.: +81-53-435-2092

**Abstract:** Diffuse optical tomography (DOT) is a biomedical imaging modality that can reconstruct hemoglobin concentration and associated oxygen saturation by using detected light passing through a biological medium. Various clinical applications of DOT such as the diagnosis of breast cancer and functional brain imaging are expected. However, it has been difficult to obtain high spatial resolution and quantification accuracy with DOT because of diffusive light propagation in biological tissues with strong scattering and absorption. In recent years, various image reconstruction algorithms have been proposed to overcome these technical problems. Moreover, with progress in related technologies, such as artificial intelligence and supercomputers, the circumstances surrounding DOT image reconstruction have changed. To support the applications of DOT image reconstruction in clinics and new entries of related technologies in DOT, we review the recent efforts in image reconstruction of DOT from the viewpoint of (i) the forward calculation process, including the radiative transfer equation and its approximations to simulate light propagation with high precision, and (ii) the optimization process, including the use of sparsity regularization and prior information to improve the spatial resolution and quantification.

**Keywords:** diffuse optical tomography; image reconstruction; inverse problem

## 1. Introduction

Diffuse optical tomography (DOT) is a biomedical imaging modality that reconstructs the distribution of the optical properties of scattering and absorption coefficients as tomographic images by employing light illumination and detection at the surface of a measured object [1–7]. Near-infrared (NIR) light propagates deep into tissues and is mainly absorbed by oxy-/deoxyhemoglobin. Therefore, by acquiring DOT with NIR light images, the concentrations of oxy-/deoxyhemoglobin and related oxygen saturation can be determined. This permits diagnosis of diseases such as breast cancer [8–12] and those involving angiogenesis, and can be used to monitor activities in the human brain [13] and to observe the oxygen supply to the neonatal brain in the intensive care unit [14], both of which are reflected by changes in regional blood flow in the brain. The scattering coefficient can be an indicator of changes in the tissue conditions.

Although DOT appears to be a promising novel biomedical imaging technology, it involves technical problems attributed to light propagation accompanied by scattering and absorption. Unlike X-ray computed tomography (CT), image reconstruction employing backprojection does not function well for DOT because of diffusive light propagation. Therefore, DOT requires image reconstruction consisting of two processes. The first is the forward process, which calculates light propagation and predicts measurements. The second is an optimization process that minimizes the error between the actual and predicted measurements.

In the forward process, light propagation can be calculated with a given set of absorption and scattering coefficients by employing the radiative transfer equation (RTE) and

various approximations of the RTE, including the photon diffusion equation (PDE), $P_N$ approximation, and Monte Carlo method. The predicted measurements at the surface of the object are obtained by solving these equations. In the optimization process, a given set of optical properties used in the forward process is updated to minimize the error between the predicted and actual measurements by employing various optimization methods such as the Newton–Raphson method, which minimizes the squared error. The use of regularization techniques and prior information is a good option in the optimization process for some applications. Various image reconstructions can be composed to solve the inverse problem by selecting and combining adequate methods for the forward and optimization processes. Because a strong scattering effect in biological tissues is essentially unavoidable, image reconstruction is crucial for realizing DOT in practical clinical use.

Studies on image reconstruction have attempted to resolve the technical problems of spatial resolution and quantification accuracy in DOT images of highly heterogeneous measured objects. Image reconstruction has been improved by the recent studies mentioned in this review, although these efforts have not been reflected in clinical applications. To overcome this low spatial resolution, sparsity regularization techniques related to compressed sensing technology have been introduced. The use of prior information about the structure inside the body obtained from other imaging modalities, such as X-ray CT and magnetic resonance imaging (MRI), has been proposed to improve spatial resolution. Spectral prior information improves the quantification of chromophore concentrations in multispectral DOT imaging. Moreover, changes in the circumstances surrounding DOT in the past two decades affect image reconstruction algorithms. Recent progress in computational technology, including artificial intelligence (AI) and high-performance supercomputers, may have altered image reconstruction of DOT. Image reconstruction schemes employing AI with deep learning in diffuse optical imaging [15–19] and the simulation of light propagation using supercomputers [20] have also been attempted in recent years. Progress in diffuse optics and related imaging technologies, including photoacoustic (PA) imaging, which allows high-resolution imaging of blood vessels deep inside the body [21,22], may promote reconsideration of the role of DOT and its image reconstruction. In such a changing situation, it is worth reviewing what has been achieved in DOT image reconstruction for research in diffuse optical imaging and related fields. This review provides useful information to select algorithms for clinical applications of DOT and will assist researchers working in emerging DOT-related research fields, including AI, high-performance computation, and different optical imaging technologies, to expand their research into DOT.

The general framework and theoretical aspects of DOT image reconstruction are described in detail in the literature [23,24]. Previous review articles [2,25–27] have provided highly informative guides for implementing DOT image reconstruction. Here, we review the methods for the forward and optimization processes with some studies on DOT image reconstruction, including some recent reports that were not included in previous reviews. The authors tried to cover as wide a range of topics and studies in this review as possible, although this may have inflated rather than simply fledged this review. Figure 1 illustrates the flow of image reconstruction for DOT and related topics for each of the processes mentioned in this review.

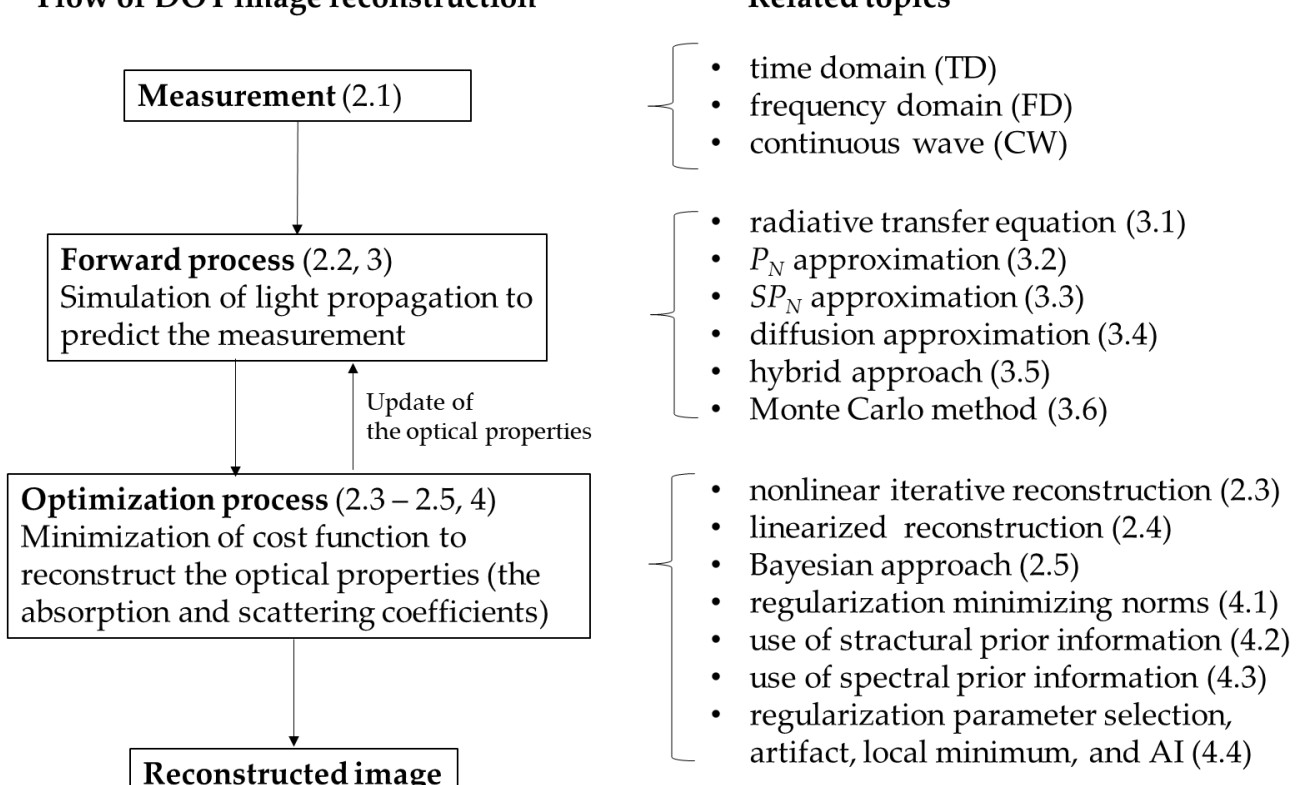

**Figure 1.** Flow of DOT image reconstruction and topics related to the processes in image reconstruction mentioned in this review together with corresponding section numbers.

## 2. Outline of DOT Image Reconstruction

### 2.1. Measurement

For image reconstruction of DOT, the distribution of optical properties, such as the absorption coefficient $\mu_a(\mathbf{r})$ and the scattering coefficient $\mu_s(\mathbf{r})$/the reduced scattering coefficient $\mu_s'(\mathbf{r})$ at position $\mathbf{r}$ in the images, must be calculated. The images of the optical properties can be translated into images of the physiological and functional properties, such as the hemoglobin concentration and oxygen saturation of blood and tissues. The optical properties are calculated by comparing the actual and predicted measurements. Therefore, image reconstruction for DOT begins with the measurement.

Detailed descriptions of the measurement for DOT image reconstruction can be found in the literature [1,3–7]. Light sources such as lasers and laser diodes are used for illumination of the surface of the medium to generate measurements of light propagating through the measured object and reaching its surface. The measurements are obtained using optical detectors that can measure the light intensity at the surface of the measured object. Optical detectors such as photomultiplier tubes connected to optical fibers on which the opposite side is attached to the measured object can be employed for the measurement. The types of measurements are usually categorized into time-domain (TD), frequency-domain (FD), and continuous-wave (CW) measurements with illumination using pulsed, intensity-modulated, and continuous light, respectively. Through TD measurement, the distribution of time-of-flight (DTOF) is obtained, and some characteristics, such as moments of DTOF (the mean time of flight of photons is the first moment of DTOF), integral transform, and the shape of the DTOF itself, can be used in image reconstruction. Through FD measurement, the intensity of light, modulation depth, and phase shift are used. The light intensity is used in the CW measurement.

### 2.2. Prediction of the Measurement: Forward Process

The forward and optimization processes are described in detail in previous studies [2,23–28]. The predicted measurements to be compared with the actual measurements are obtained by computing the light measured at the surface of the object based on the theory of light propagation inside biological media. Light propagation involves the transport of photons, accompanied by scattering and absorption. In this case, the radiative transport equation (RTE) describes the light propagation and can be used to obtain predicted measurements with a set of given optical properties. Some approximation methods of the RTE, such as $P_N$ approximation, diffusion approximation (DA) with the photon diffusion equation (PDE), and the Monte Carlo (MC) method are used in the computation of light propagation to reduce the computational difficulties of the RTE. Except for the MC method, general numerical computation methods such as the finite element method (FEM) and the finite difference method are used to compute light propagation and predicted measurements.

### 2.3. Optimization Process for Nonlinear Image Reconstruction

Let $F_{i,j}(\mu_a, \mu_s{}')$ be the mathematical operation for calculating the measured light with optical properties $\mu_a$ and $\mu_s{}'$ and the given positions of the light sources and detectors to obtain the predicted measurements: $\mu_s{}' = (1 - g)\mu_s$ is defined by the anisotropy (asymmetry) factor, $g$, which is the average cosine of the scattering angle. Image reconstruction can be formulated as an optimization problem to find $\mu_a$ and $\mu_s$ while minimizing the cost function, which consists of the squared error and additive regularization term, as follows:

$$\min_{\mu_a, \, \mu_s{}'} \left\{ \frac{1}{2} \sum_{i=1}^{I} \sum_{j=1}^{J} w_{i,j} \big( F_{i,j}(\mu_a, \, \mu_s{}') - M_{i,j} \big)^2 + \gamma \cdot R(\mu_a, \, \mu_s{}') \right\}, \tag{1}$$

where $M_{i,j}$ are the measurements with the $i$th source and $j$th detector, respectively, and $w_{i,j}$ is the weight adjusting the contribution of $M_{i,j}$ to the image reconstruction. $R(\mu_a, \mu_s{}')$ is a regularization term. The image-reconstruction process used to solve the inverse problem is generally ill-posed. Image reconstruction often suffers from the nonuniqueness of solutions and instability owing to noise contaminating the measurements. $R(\mu_a, \mu_s{}')$ is a function for evaluating the extent to which the solution (reconstructed image) satisfies the condition expected in the desired solution and for reducing the overfitting of $F_{i,j}$ to $M_{i,j}$ contaminated by noise. By minimizing $R(\mu_a, \mu_s{}')$ with a residual error, this ill-posed nature can be alleviated. Various types of $R(\mu_a, \mu_s{}')$, such as the squared $L_2$-norm of the reconstructed image (Tikhonov regularization), $L_p$-norm ($0 \leq p \leq 1$), and total variation, which is the norm of the gradient of the image, can be applied. $g$ adjusts the regularization effect.

The relationship between the optical properties and measurements as a result of light propagation is nonlinear. Therefore, image reconstruction is performed by solving Equation (1) by employing nonlinear optimization methods with iterative updating processes, such as the Newton–Raphson, quasi-Newton, and conjugate gradient methods. For the nonlinear optimization process, the gradient of the cost function $f$, the sum of the squared error term, with the regularization term in Equation (1) must be computed, as follows:

$$\frac{\partial f}{\partial \mu_k} = \sum_{i=1}^{I} \sum_{j=1}^{J} w_{i,j} \big( F_{i,j}(\mu_k) - M_{i,j} \big) \frac{\partial F_{i,j}}{\partial \mu_k} + \gamma \cdot \frac{\partial R(\mu_k)}{\partial \mu_k}, \tag{2}$$

where $\mu_k$ is the optical property of interest at the $k$th position $\mathbf{r}_k$, where one of the nodes for the FEM to compute $F_{i,j}$ exists. The vector matrix formula in Equation (2) is often used.

$$\nabla f(\boldsymbol{\mu}) = \frac{\partial f}{\partial \boldsymbol{\mu}} = W J^T (\mathbf{F} - \mathbf{M}) + \gamma \cdot \frac{\partial R}{\partial \boldsymbol{\mu}}, \tag{3}$$

where $\mathbf{F}$ and $\mathbf{M}$ are column vectors in which $F_{i,j}$ and $M_{i,j}$ are aligned, respectively. $\boldsymbol{\mu}$ is a vector of $\mu_k$. $W$ is the diagonal matrix with $w_{i,j}$, and $J$ is the Jacobian matrix with $\partial F_{i,j}/\partial \mu_k$ as its elements, which can be calculated by the perturbation method or the adjoint method

derived in the early important work on the DOT image reconstruction by Arridge and Schweiger [23,28]. Using the Newton–Raphson method, iterative image reconstruction can be expressed as

$$\boldsymbol{\mu}_{t+1} = \boldsymbol{\mu}_t - H^{-1}(\boldsymbol{\mu}_t)\nabla f(\boldsymbol{\mu}_t), \tag{4}$$

where $H = \nabla^2 f(\boldsymbol{\mu})$ is a Hessian matrix and the subscript $t$ is the number of iterations. The updating process in Equation (4) is repeated until the solution converges.

The Levenberg–Marquardt method, which is often employed in DOT image reconstruction, uses the following iteration to solve Equation (2), with $W$ equal to an identity matrix and $\gamma = 0$.

$$\boldsymbol{\mu}_{t+1} = \boldsymbol{\mu}_t - (J^T J + \gamma' \cdot I)\nabla f(\boldsymbol{\mu}_t), \tag{5}$$

where $\gamma'$ is a small positive value and $I$ an identity matrix. The nonlinear optimization methods, such as the steepest descent and conjugate gradient methods, have been compared for DOT image reconstruction with DA [29].

### 2.4. Optimization Process for Linearized Image Reconstruction

This section describes the optimization process for image reconstruction using a linearized forward process. As a good approximation, the $\overline{\boldsymbol{\mu}}$ of the reconstructed image $\boldsymbol{\mu}$ is obtained, and the measurements are approximated using the Taylor expansion as

$$\mathbf{M}(\boldsymbol{\mu}) = \mathbf{F}(\overline{\boldsymbol{\mu}}) + J(\overline{\boldsymbol{\mu}})\delta\boldsymbol{\mu} + \cdots, \tag{6}$$

where $\mathbf{F}(\overline{\boldsymbol{\mu}})$ is the predicted measurement with $\overline{\boldsymbol{\mu}}$ (or the measurement equivalent to the predicted measurement) and $\delta\boldsymbol{\mu} = \boldsymbol{\mu} - \overline{\boldsymbol{\mu}}$. When the series expansion is terminated in the first order, the forward process can be linearized, and the linear equation

$$(\mathbf{M} - \mathbf{F}) = J\delta\boldsymbol{\mu}, \tag{7}$$

is obtained. The linearizations of the forward process for the absolute and logarithmic values of the measured light intensities are referred to as the Born and Rytov approximations, respectively. Then, the optimization process can be formulated in a manner similar to that in Equation (1) with regularization, as follows:

$$\min_{\delta\boldsymbol{\mu}}\left\{(\delta\mathbf{M} - J\delta\boldsymbol{\mu})^T W(\delta\mathbf{M} - J\delta\boldsymbol{\mu}) + \lambda R(\delta\boldsymbol{\mu})\right\}, \tag{8}$$

where $\delta\mathbf{M} = \mathbf{M} - \mathbf{F}$. When employing classical Tikhonov regularization with a weighting matrix $L$ formulated as $R(\delta\boldsymbol{\mu}) = \|L\delta\boldsymbol{\mu}\|^2$, $\delta\boldsymbol{\mu}$ is obtained by setting the derivative of the objective function of Equation (7) to zero,

$$\delta\boldsymbol{\mu} = \left(J^T W J + \gamma \cdot L^T L\right)^{-1} J^T W\delta\mathbf{M}, \tag{9}$$

and the reconstructed image is expressed as $\boldsymbol{\mu} = \overline{\boldsymbol{\mu}} + \delta\boldsymbol{\mu}$. Depending on the type of regularization term, a nonlinear optimization process is needed to solve Equation (7). The limitations of linearized image reconstruction have been previously discussed [30]. The linearization approach is appropriate when changes in the optical properties are sufficiently small and exist in small regions.

### 2.5. Bayesian Approach

The measurements are always contaminated with additive random noise. Thus, the forward equation with a given optical property can be written as $\mathbf{M} = \mathbf{F}(\mu) + \varepsilon$, where $\varepsilon$ is the random additive noise and $\mu$ is the given optical property. Owing to the randomness of $\varepsilon$, $\mathbf{M}$ is a random variable with a conditional (prior) probability density function $p(\mathbf{M} \mid \mu)$. Assuming $\mu$ and $\varepsilon$ are stochastically independent, the joint probability is formulated as $p(\mathbf{M}, \mu) = p(\mathbf{M} \mid \mu) \cdot p(\mu) = p(\mu \mid \mathbf{M}) \cdot p(\mathbf{M})$, which is Bayes' rule.

In the Bayesian approach based on the above formula, image reconstruction is to find $\mu$ maximizing the conditional (posterior) function, $p(\mu \,|\, \mathbf{M}) = p(\mathbf{M} \,|\, \mu) \cdot p(\mu)$ as a likelihood function under the condition of $\mathbf{M}$ determined, i.e., $p(\mathbf{M}) = 1$. When $\varepsilon$ is Gaussian random noise with a covariance matrix $W$ and an average of zero, the log-likelihood of the posterior function is equivalent to Equation (1) with $p(\mu) = \exp(-\lambda \cdot R(\mu))$. Assuming that $\mu$ is a Gaussian random variable with a variance $1/\lambda$, the image reconstruction incorporates classical Tikhonov regularization.

By choosing the probability density function, the Bayesian approach can take advantage of regularization methods and prior knowledge for image reconstruction, as shown in previous studies [31–33]; some Bayesian approaches are introduced in the following sections.

The image reconstruction method can be characterized by the methods selected for the forward and optimization processes, particularly the equations employed in the forward process and the cost functions with a certain regularization and use of prior information in the optimization process. Several studies on image reconstruction algorithms are categorized and introduced in the following sections from the viewpoint of the forward and optimization processes.

## 3. Forward Process

### 3.1. Radiative Transfer Equation

Light propagation through biological tissues is mathematically described by the radiative transfer equation (RTE), as follows [23]:

$$\left\{ \frac{1}{c} \frac{\partial}{\partial t} + \mathbf{s} \cdot \nabla + \mu_a(\mathbf{r}) + \mu_s(\mathbf{r}) \right\} I(\mathbf{r}, \mathbf{s}, t) = \mu_s(\mathbf{r}) \int_{4\pi} p(\mathbf{s} \cdot \mathbf{s}') I(\mathbf{r}, \mathbf{s}', t) d\Omega' + q(\mathbf{r}, \mathbf{s}, t), \quad (10)$$

where $c$ is the speed of light, $I(\mathbf{r}, \mathbf{s}, t)$ the radiance (light intensity), and $q(\mathbf{r}, \mathbf{s}, t)$ the light source term. $\mathbf{r}$ and $\mathbf{s}$ represent the position and direction of light, respectively. $p(\mathbf{s} \cdot \mathbf{s}')$ is the scattering phase function, which represents the probability that the direction of light $\mathbf{s}'$ is changed to $\mathbf{s}$ by the scattering event. The Henyey–Greenstein function is commonly used as $p(\mathbf{s}, \mathbf{s}')$ in DOT image reconstruction. Equation (10) represents the time-dependent case. The time-derivative term on the left side of Equation (10) is removed in the case of time-independent CW measurement. Fourier-transformed versions of Equation (10) are used for FD measurements.

The computation of the RTE does not impose any limitations on the optical properties. Therefore, RTE can compute light propagation more accurately than other methods, such as DA, and can be the best choice for image reconstruction for objects that include void-like regions, such as the trachea in the neck, which can affect DOT imaging of thyroid cancer [34,35].

Klose and Hielscher employed time-independent RTE for image reconstruction [36,37]. For the computation of the RTE, they employed the upwind-difference discrete-ordinate method by discretizing the angular and spatial variables in the RTE. After experimental validation of the forward process with a rectangular phantom, including the void-like region [36], image reconstruction using the gradient method, referred to as model-based iterative image reconstruction (MOBIIR), was performed. Images of $\mu_a$ and $\mu_s$ were obtained in a phantom containing void-like regions [37]. Abdoulaev also reported MOBIIR with a forward process using FEM [38].

Tarvainen et al. used the RTE in FD measurements during the forward process [39]. The RTE was computed using the FEM, in which stream diffusion modification was utilized. In the optimization process, the cost function with total variation regularization (see Section 4.1) was minimized using the Gaussian–Newton method. Image reconstruction was successful in the 2D numerical simulations of various cases, including low-scattering blobs and low-scattering and low-absorption gaps.

Soloviev and Arridge proposed RTE-based image reconstruction for the FD measurement of an object, such as an embryo that consists of weakly and highly scattering

regions [40]. Assuming that $\mu_a + \mu_s$ is significantly different between the weak and high scattering regions, an approximation of the RTE solution was derived for the forward process. The optimization process with Tikhonov regularization was demonstrated using 3D numerical simulations.

Machida et al. proposed image reconstruction with linearization employing the Rytov approximation accompanied by Green's function calculated using the $F_N$ method [41]. The $F_N$ method was originally used to compute the 1D RTE using singular-eigenfunction expansion, which was extended to 3D computation by employing the rotated reference frame method [42]. The image reconstruction method was examined in phantom experiments using CCD measurements. Machida also reported numerical simulations of image reconstruction based on the Born approximation and 3D FN computation of the RTE for a slab geometry with spatially oscillating structured light [43].

Image reconstruction employing RTE is particularly suitable for small measurement objects such as finger joints. Clinical applications for diagnosing rheumatoid arthritis have been reported [44–46]. Image reconstructions based on the RTE have been reported in fluorescent diffuse optical tomography (FDOT) [47], which reconstructs the concentrations of fluorescent probes [48,49], and in quantitative photoacoustic tomography (QPAT) [50], which reconstructs the concentrations of exogenous and endogenous chromophores from photoacoustic pressure waves [51–54]. FDOT and QPAT are used to image subjects that exist in shallow regions and in small objects, where DA is often invalid. Efficient and precise computational methods for RTE have been studied [20,55].

### 3.2. $P_N$ Approximation

To obtain a good approximation of the RTE, $P_N$ approximation [23] was attempted in the forward process of image reconstruction. The $P_N$ approximation is derived from an orthogonal function expansion using spherical harmonics in Equation (10), in which $I(\mathbf{r},\mathbf{s},t)$, $q(\mathbf{r},\mathbf{s},t)$, and $p(\mathbf{s}\cdot\mathbf{s}')$ [23] are formulated as

$$I(\mathbf{r},\mathbf{s},t) = \sum_{l=1}^{\infty} \sum_{m=-l}^{l} \left(\frac{2l+1}{4\pi}\right)^{\frac{1}{2}} \psi_{l,m}(\mathbf{r},t) Y_{l,m}(\mathbf{s}), \tag{11}$$

$$q(\mathbf{r},\mathbf{s},t) = \sum_{l=1}^{\infty} \sum_{m=-l}^{l} \left(\frac{2l+1}{4\pi}\right)^{\frac{1}{2}} q_{l,m}(\mathbf{r},t) Y_{l,m}(\mathbf{s}), \tag{12}$$

$$P(\mathbf{s} \cdot \mathbf{s}') = \sum_{l=1}^{\infty} \sum_{m=-l}^{l} \Theta_{l,m} Y_{l,m}^*(\mathbf{s}') Y_{l,m}(\mathbf{s}), \tag{13}$$

where $\psi_{l,m}$, $q_{l,m}$, and $\Theta_{l,m}$ are the coefficients, and $Y_{l,m}$ the spherical harmonics of order $l$ and degree $m$, which are associated with the Legendre polynomial and orthonormal function. $Y_{l,m}^*$ is the complex conjugate of $Y_{l,m}$. Owing to the orthonormality of the spherical harmonics, the RTE, in which Equations (11)–(13) are substituted, is decoupled into $(N + 1)!$ simultaneous equations by multiplying $Y_{l,m}^*$ when the spherical harmonic expansion is terminated at $l = N$. The $P_N$ approximation is computed with simultaneous equations.

Boas et al. compared $P_3$ approximation with DA for the determination of the optical properties from MC simulated measurements and from $P_3$-approximated measurements in the FD measurement [56]. $\mu_a$ was more accurately determined using $P_3$ approximation, although the DA estimated $\mu_s'$ better than $P_3$ approximation in the highly forward-scattering case. Jiang and Paulsen derived a stable and computationally convenient higher-order DA ($P_3$ approximation) with second-order spatial derivative terms and compared the measurements with a cylindrical phantom and the computational results with the higher-order DA and DA using FEM [57].

Oliveria and Tahir developed a computer program based on their original code, called EVENT, to compute the $P_N$ approximation with FEM and compared the computational results with the $P_N$ approximation and with DA. They succeeded in reconstructing

images from MC-simulated measurements by employing $P_7$ approximation in TD measurements [58].

Jiang developed an image reconstruction algorithm based on the third-order diffusion approximation ($P_3$ approximation) with Marquardt and Tikhonov regularizations and succeeded in image reconstruction using 2D numerical simulations [59]. Yuan et al. implemented a 3D FEM computation of $P_3$ approximation and compared $P_3$ approximation with $P_1$ approximation and MC simulation. They reconstructed an image of a 3D numerical phantom mimicking a finger joint with cartilage between the bones [60].

Wright et al. performed image reconstruction implementing the FEM calculation of the $P_N$ approximation in the forward process in FD measurement [61]. They employed the even-parity formulation of RTE, which was derived by introducing the light intensity, $I^{\pm}(\mathbf{s}) = \{I(\mathbf{s}) \pm I(-\mathbf{s})\}/2$, and the light source, $q^{\pm}(\mathbf{s}) = \{q(\mathbf{s}) \pm q(-\mathbf{s})\}/2$. Consequently, the RTE was transformed into the formula of $I^+$ including the second-order spatial derivative. The simultaneous equations of $P_N$ approximation, which are expressed in the vector matrix formula, are discretized using the FEM. Numerical simulations of image reconstruction employing the $P_1$, $P_3$, and $P_5$ approximations were performed, with the measurements computed using the $P_7$ approximation. It was demonstrated that the highly absorbing and scattering regions in the low-scattering medium were imaged more accurately in the reconstructed image using the higher-order $P_N$ approximation.

*3.3. $SP_N$ Approximation*

Klose and Larsen introduced a simplified spherical harmonics ($SP_N$) approximation, which was originally proposed for neutron transport, for CW light propagation in biological media [62]. In the $SP_N$ approximation, the $P_N$ approximation of 1D light propagation in the planar geometry was directly applied to 3D light propagation by replacing the 1D differential operator, $\mathrm{d}/\mathrm{d}x$, by the 3D gradient operator, $\nabla$. In the 1D $P_N$ approximation, the Legendre moments of radiance $\phi_n(x) = \int P_n(\hat{s}) I(x, \hat{s}) d\Omega$ are defined. A set of four equations for the $SP_7$ approximation was formulated for composite moments $\varphi_m$ ($m = 1$, 2, 3, 4) defined by the sum of the series of $\varphi_n$ ($n = 1, 2, \ldots, 6$). The equations for the $SP_N$ approximation ($N < 7$) were readily derived from those for the $SP_7$ approximation, and the equation for $SP_1$ approximation was equivalent to the PDE described in Section 3.4. The $SP_N$ approximation was calculated using a set of coupled diffusion-like equations with second-order spatial derivative terms. The numerical simulations in [62] indicated that the precision of the approximation improved rapidly from $N = 1$ to 7. The computational time of the $SP_N$ approximation is significantly shorter than that for the RTE computation by the discrete ordinate method discretizing $\mathbf{s}$.

Chu et al. developed a 3D FEM for $SP_N$ approximation in FD measurements [63]. The computational results were compared with those of MC simulations. The $SP_N$ approximation ($N > 1$) provides a more accurate phase and amplitude of the modulated light than $SP_1$ approximation. Chu and Dehghani reported image reconstruction in FD measurements using the $SP_7$ approximation [64]. The 2D image reconstructions of $\mu_a$ and $\mu_s$ from the phase and amplitude obtained by the numerical simulation with the $SP_5$ approximation were performed by employing the Jacobian matrix, $J$, calculated using the $SP_1$ and $SP_5$ approximations. Image reconstruction using the $SP_5$ approximation provided more accurate quantitative and qualitative images.

Domínguez and Bérubé-Lauzière reported image reconstruction using time-domain parabolic $SP_N$ equations [65,66]. In addition to the aforementioned CW and FD cases, a system of diffusion-like equations for the $SP_7$ approximation was formulated for the TD measurement. Numerical simulations of image reconstruction employing $N = 1, 3, 5$, and 7 were performed. In the simultaneous reconstruction of the heterogeneous medium where the high-absorption inclusion and high-scattering inclusion existed, the reconstructed $\mu_a$ and diffusion coefficient, $D = 1/\{3(\mu_a + \mu_s')\}$, were evaluated. The $SP_N$ approximations ($N > 1$) reconstructed the high $\mu_a$ inclusion and low $D$ inclusions better than the $SP_1$

approximation, which underestimated $\mu_a$. The error in the reconstructed value with the $SP_7$ approximation was slightly larger than that with the $SP_5$ and $SP_5$ approximations [66].

Some studies on image reconstruction with $SP_N$ approximations for FDOT and QPAT have been reported [67–69].

### 3.4. Diffusion Approximation

In DA, light propagation in the TD measurement is described by the following PDE [23]:

$$\left\{\frac{1}{c}\frac{\partial}{\partial t} + \mu_a(\mathbf{r}) - \nabla \cdot D(\mathbf{r})\nabla\right\}\Phi(\mathbf{r}, t) = q_0(\mathbf{r}, t), \tag{14}$$

where $D = 1/\{3(\mu_a + \mu_s')\}$ is the diffusion coefficient, $\Phi(\mathbf{r}, t) = \int_{4\pi} I(\mathbf{r}, \mathbf{s}, t)d\Omega$ the fluence rate (photon density), and $q_0(\mathbf{r}, t)$ an isotropic light source. For the CW measurement, the first term in Equation (14) is removed. Equation (14) is the Fourier transform for the FD measurement.

To compute light propagation with the PDE, the Robin boundary condition, which is often used in image reconstruction, is as follows:

$$\frac{1}{2A}\Phi(\mathbf{r}, t) = -\mathbf{n} \cdot D(\mathbf{r})\nabla\Phi(\mathbf{r}, t), \tag{15}$$

where $A$ is a parameter that depends on the internal reflection ratio, and $\mathbf{n}$ is the outward normal vector. Early studies by Arridge and Schweiger [28] established the standard nonlinear image reconstruction scheme for DOT. The computation of the PDE is typically performed using the FEM. The Jacobian matrix $J$ appearing in Equation (3) is obtained using the adjoint method based on the reciprocity theorem [70,71]. In the TD measurement, the perturbation $\eta$ in $\Phi$ with $\mu_s'$ and $\mu_a$ at the measured position $\mathbf{r}_d$ associated with small changes $\nu$ and $\alpha$ in $\mu_s'$ and $\mu_a$, respectively, is calculated as

$$\eta(\mathbf{r}_d, t) = -\int_\Omega \int_{-\infty}^\infty \nu(\xi)\nabla\Psi(\mathbf{r}_d, \xi, t, \tau) \cdot \nabla\Phi(\xi, \tau) + \alpha(\xi)\Psi(\mathbf{r}_d, \xi, t, \tau)\Phi(\xi, \tau)d\tau d\xi, \tag{16}$$

where $\Psi$ is Green's function of the adjoint PDE, which calculates the light propagation from $\mathbf{r}_d$ with position $\xi$ and time $\tau$. From Equation (16), the entries of $J$ with respect to $\mu_s'$ are computed with the inner product of the gradients of $\Psi$ and $\Phi$, discretized with the FEM, whereas those with respect to $\mu_a$ are computed with the product of $\Psi$ and $\Phi$ on the right-hand side of Equation (16).

Various types of measurement exist for image reconstruction. Gao et al. compared reconstructed images from the measured intensity (CW measurement), mean time-of-flight (first moment), variance (second moment), skew (third moment), and full-time-resolved DTOF [72]. The normalized DTOF provides a better image in terms of spatial resolution and quantification in 2D numerical experiments using DA, although the image reconstruction from full-time-resolved measurements required a 120-fold longer computational time than that from the mean time-of-flight and variance. DA image reconstruction from Laplace-transformed DTOF using a modified generalized pulse spectrum technique (mGPST) has also been proposed [73].

DA is obtained from the $P_1$ approximation with the assumption that the time derivative of the net flux vector, $\mathbf{J} = \int_{4\pi} \mathbf{s}I(\mathbf{r}, \mathbf{s}, t)d\Omega$, equals zero, and the light source is isotropic. The DA is valid while $\mu_a << \mu_s'$. It is not valid to use the PDE for the nonscattering void region. Moreover, DA requires spatiotemporal conditions in which the directions of photons are sufficiently randomized by many scattering events and the light intensity changes very slowly [23,74].

Although the limitation of DA requiring the abovementioned assumptions and conditions, which often seem difficult to be fulfilled, has been debated, image reconstruction employing PDE has been successfully applied in in vivo and clinical DOT studies [75–79]. Several studies on image reconstruction based on DA with various regularizations and

inversion schemes have been reported because the computational burden of the PDE is reasonable. Most studies on the optimization process introduced in Section 4 employed DA.

Sophisticated open sources for the forward computation of light propagation and image reconstruction, such as TOAST++ [80,81] and NIRFAST [82,83], are currently publicly available and have contributed to countless studies on DOT applications and image reconstruction.

### 3.5. Hybrid Approach

Although some limitations exist, DA can significantly reduce the computational burden compared with the RTE computation. Taking advantage of DA, hybrid methods of RTE and DA have been studied [84–86]. The RTE is used only in the spatial and temporal domains, wherein using DA can be invalid for hybrid approaches.

Tarvainen et al. succeeded in image reconstruction by employing a forward process coupling the RTE and PDE computations in 2D numerical simulations of the FD measurement [87]. The measured circular object was divided into two circular subdomains. The RTE was used in one of the subdomains near the measured surface, whereas the PDE was used in the inner subdomain. The forward calculation coupling the RTE and PDE was implemented using FEM. The optimization problem with weighted Tikhonov regularization was solved using the Gauss–Newton method. It was demonstrated that the coupled RTE and DA models reconstructed low-scattering inclusions as well as the RTE model from the measurements simulated using the MC method, including Gaussian-distributed noise.

### 3.6. Monte Carlo Simulation

Generally, the MC method is a numerical technique used to solve deterministic problems such as partial differential equations using probabilistic methods [88,89]. In the forward process of computing light propagation, the movements of photons with scattering and absorption by biological tissues were stochastically simulated. Monte Carlo modeling of light transport in multilayered tissues (MCML) developed by Wang et al. and other software available online are widely used [90–93]. In the MC method, a photon packet with unit energy $w$ is launched into a medium comprising layers with optical properties. The length of the free path in which the photon packet is not scattered is determined by an exponentially distributed random variable. The ratio of photons maintaining the direction and surviving the absorption at distance $l$ is given by $p(l) = \mu_t \exp(-\mu_t l)$, where $\mu_t = \mu_a + \mu_s$. The energy of the photon packet decreases with absorption at the position reached by the photon packet. The absorbed energy $(\mu_a/\mu_t)w$ is recorded at this position. The photon packet with the remaining energy $(\mu_s/\mu_t)w$ continues to travel. The change in the direction of the photon packet by a scattering event is determined randomly using the Henry–Greenstein function $p(\mathbf{s},\mathbf{s}')$. The photon packet is reflected randomly by obeying Fresnel reflectance at the boundaries of the layers. A large number of photon packets are traced such that the recorded absorbed light distribution converges to a good approximation of the light propagation. Comparisons between MC and DA have been reported in many studies [94–97].

Hayakawa et al. proposed the perturbation MC (pMC) method for the computation of the Jacobian matrix $J$ to solve the inverse problem using a gradient-based nonlinear optimization process [98]. In the pMC method, the standard MC method is first used to record the energies of the detected photon packets moving from the light source to detectors with scattering events. To compute the changes in the measurements, the recorded energy $w$ of the photon traveling through the region with the perturbations of $\mu_s \to \overline{\mu}_s$, $\mu_a \to \overline{\mu}_a$, and $\mu_t \to \overline{\mu}_t$ is modified as

$$\overline{w} = w\left(\frac{\overline{\mu}_s/\overline{\mu}_t}{\mu_s/\mu_t}\right)^{\zeta}\left(\frac{\overline{\mu}_t}{\mu_t}\right)^{\zeta}\exp\{-(\overline{\mu}_t - \mu_t)L\}, \tag{17}$$

where $L$ is the total length of the photon packet traveling through the region perturbed by $\zeta$ scattering events. The matrix $J$ can be obtained using Equation (17). The performance

of the pMC method was numerically examined to determine the optical properties of an object with two layers.

Kumar and Vasu demonstrated image reconstruction for a heterogeneous medium with the background and inclusions having a low $\mu_s$ of approximately 0.5 mm$^{-1}$, because of which the DA did not hold [99]. A nonlinear iterative optimization method, called the conjugate gradient squared method, was employed. Matrix $J$ was computed using the pMC method. $\mu_a$ and $\mu_s$ values of the inclusions were simultaneously reconstructed with/without prior information regarding the locations of the inclusions in the 2D numerical simulations. Image reconstruction using the pMC method succeeded in imaging inclusions that could not be reconstructed using DA.

Boas et al. simulated light propagation in a 3D realistic adult head model using the MC method [100]. Then, using the Rytov approximation, which is a series expansion of the logarithm of $\Phi$, the linearized forward process was formulated, as shown in Equation (7). In this study, the entries of matrix $J$ to reconstruct the changes in $\mu_a$ from the Rytov approximation were calculated using the MC method based on the adjoint method with a light source with the $\delta$-function in the CW version, as follows [101]:

$$J_{m,n} = \frac{G(\mathbf{r}_{s,m}, \mathbf{r}_n)G(\mathbf{r}_n, \mathbf{r}_{d,m})}{G(\mathbf{r}_{s,m}, \mathbf{r}_{d,m})}, \tag{18}$$

where $G(\mathbf{r}_1, \mathbf{r}_2)$ is Green's function, which is the fluence rate at $\mathbf{r}_2$ generated by the light source at $\mathbf{r}_1$; $\mathbf{r}_{s,m}$ and $\mathbf{r}_{d,m}$ are the positions of the $m$th pair of the source and detector, respectively. $\mathbf{r}_n$ is the $n$-th voxel that discretizes the medium to the source positions. By the image reconstruction using Equation (18) and standard Tikhonov regularization, the changes in regional blood flow in the somatosensory area during median nerve stimulation were imaged and superimposed on the magnetic resonance image of the subject's head.

## 4. Optimization Process

### 4.1. Use of Regularization Minimizing Norms

Regularization methods were employed in DOT image reconstruction to solve the inverse problem and alleviate instability, which means that reconstruction is strongly affected by noise in the measurement and error in the forward process. The reconstructed image was disturbed by noise and errors. As shown in Equation (9), the classical Tikhonov regularization that minimizes the squared 2-norm ($\ell_2$-norm or $L_2$-norm) of the reconstructed image can be readily applied to image reconstruction. Tikhonov regularization with an appropriate $\gamma$ is useful for obtaining a smooth image by reducing the effect of measurement noise. However, smoothness is often accompanied by a decline in spatial resolution. In such a case, the changes in the optical properties of the measured object are blurred and are reconstructed in a larger volume than the true one. As a result, the change is reconstructed as a smaller value than true one, which means that the concentration of the photon absorber such as hemoglobin is underestimated by image reconstruction with the Tikhonov reconstruction. Additionally, the effect of diffusive light propagation also causes a low spatial resolution, generally in the image reconstruction of optical imaging, as reported in the literature [102].

To improve the quality of the reconstructed image, the regularization method for minimizing the $p$-norm ($0 \leq p \leq 1$) of the reconstructed image has been used in recent years. The $p$-norm is defined as $\|\boldsymbol{\mu}\|_p = \left(\sum_{k=1}^{K}|\mu_k|^p\right)^{1/p}$, for the vector $\boldsymbol{\mu}$ comprising $K$ entries. Figure 2 illustrates the idea of the regularization minimizing $p$-norm [103]. The dashed line represents the set of optical properties $\boldsymbol{\mu} = (\mu_1, \mu_2)$ at two positions which provide an identical and single measurement. The image reconstruction of two optical properties from a single measurement is underdetermined. By employing a certain value of the norm of the reconstructed image (solid line), the number of candidates of the solution can be reduced. The candidates exist on intersecting points of the dashed line and circle, which represent the set of $\boldsymbol{\mu}$ with a certain value norm in the 2-norm case. By choosing smaller $p$,

it is observed that the intersection points approach one or the other of the axes. Thus, one of entries of $\mu$ at a position takes a large value, and the other takes a value close to zero. This indicates that the solution becomes sparse. Generally, the DOT image reconstruction has a large number of unknown changes in $\mu_a$ and $\mu_s$, which are localized in a smaller number of voxels (smaller region) with smaller value of $p$ as well as in the case of Figure 2. Then, the blurriness of the image can be reduced. In the case of Figure 2, by minimizing the $p$-norm, the solution can be specified on the tangent point. A difficulty encountered while implementing the regularization minimizing $p$-norm is computing the gradient for the optimization because the cost function becomes nonconvex. There are several techniques to implement the regularization minimizing $p$-norm.

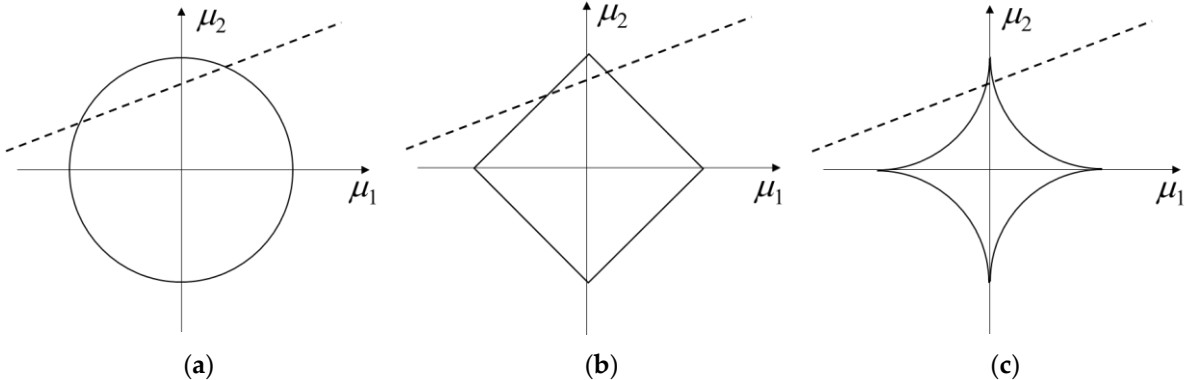

**Figure 2.** Idea of the regularization minimizing $p$-norm of the reconstructed image with (**a**) $p = 2$ (Tikhonov regularization), (**b**) $p = 1$, and (**c**) $0 < p < 1$. The solid lines represent the points of ($\mu_1$, $\mu_2$) with a constant value of the $p$-norm. The dashed lines represent the group of points providing identical measurements. The cross-section points of the solid and dashed lines are solutions with a certain value of the $p$-norm. By minimizing the norms, the tangent points are selected as the optimal reconstructed image.

Süzen et al. proposed a compressed sensing method for DOT [104]. In this study, the DOT images were reconstructed using measurements acquired via random sampling. Image reconstructions were performed using the regularization method, minimizing the $L_1$ norm (1-norm) of the image. The optimization process was based on the literature [105], in which the $L_1$-norm, i.e., the absolute value, was approximated as $|\mu| \approx \sqrt{\mu \cdot \mu + u}$, with parameter $u$ smoothing the function, and the gradient was approximately computed as $d|\mu|/d\mu \approx \mu/\sqrt{\mu \cdot \mu + u}$. $L_1$-norm regularization was found to be more robust than Tikhonov ($L_2$-norm) regularization in terms of reducing the number of measurement samples used in image reconstruction.

Shaw and Yalavarthy proposed image reconstruction with dynamic CW-domain DOT that employed image reconstruction with Rytov approximation for video-rate imaging [106]. They applied the $\ell_1$-norm (1-norm) minimization implemented with the open-source YELL1 algorithm [107] derived from the alternating direction method (ADM), which minimizes the augmented Lagrangian function equivalent to Equation (8)

$$L(\delta\mu, \mathbf{e}, \Gamma) = \|\delta\mu\|_1 + \frac{1}{2\gamma_1}\|\mathbf{e}\|_2^2 - \Gamma^T(\delta\mathbf{M} + \mathbf{e} - J\delta\mu) + \frac{\gamma_2}{2}\|\delta\mathbf{M} + \mathbf{e} - J\delta\mu\|_2^2, \quad (19)$$

by updating sequentially the optical properties $\delta\mu$, the residual error norm $\mathbf{e}$, and the Lagrange multiplier $\Gamma$ [108]. The update of $\delta\mu$ was performed using the product of the absolute value and the sign of the gradient with a threshold for minimizing the $\ell_1$-norm. Phantom experiments demonstrated that $\ell_1$-norm regularization improved the image contrast compared to traditional $\ell_2$-norm regularization.

Kavuri et al. reported that $L_1$-norm regularization improved DOT image reconstruction in terms of spatial resolution and depth localization in phantom experiments [109].

$L_1$-norm regularization was implemented using the depth-compensation method. The Rytov approximation was employed to linearize image reconstruction. In this study, matrix $M$ for depth compensation was computed using the singular value decomposition of matrix $J$. Then, matrix $J$ in Equation (7) is replaced by $JM$. $L_1$-norm regularization was implemented with a logarithmic barrier penalty function bounding the reconstructed values at a certain interval, which was minimized simultaneously with the cost function.

Cao et al. implemented sparsity regularization with $L_1$-norm by employing an expectation minimization (EM) algorithm [110]. They modeled the measurements (Rytov approximation) using additive noise and optical properties as random Gaussian variables. Subsequently, a Bayesian framework was applied for image reconstruction. The cost function was formulated as the log-likelihood function of the posterior probability to be maximized, which is equivalent to the minimization in Equation (1), where matrix $W$ is composed of the variance in the Gaussian distribution of the measurement additive noise, and the regularization term is the $L_1$-norm of the optical properties. The expectation-maximization (EM) algorithm consisted of E- and M-steps. In the E-step, the log-likelihood (cost function) is computed using the measurements and optical properties in each iteration. In the M-step, the log-likelihood is maximized. The soft threshold method was used, in which the gradient of the $L_1$-norim was computed as $\mathrm{sgn}(\delta\mu)|\delta\mu|$ and as zero when $|\delta\mu|$ was smaller than the threshold.

Okawa et al. used the $\ell_p$-norm ($0 < p \leq 1$) of a reconstructed image as a regularization term using the focal underdetermined system solver (FOCUSS) algorithm [111,112]. The change in the absorption coefficient $\delta\mu_a$ from the background was formulated to differentiate the $\ell_p$-norm as $\delta\mu_{ak} = \mathrm{sgn}(z_k)|z_k|^{2/p}$ at the $k$-th FEM node with parameter $z_k$. Then, the optimization problem of the image reconstruction in Equation (1) was reformulated as follows:

$$\min_{\mathbf{z}} \left\{ \sum_{i=1}^{I} \sum_{j=1}^{J} \left( F_{i,j}(\mathbf{z}) - M_{i,j} \right)^2 + \gamma \cdot \sum_{k=1}^{K} |z_k|^2 \right\}, \tag{20}$$

where $\mathbf{z}$ is a vector composed of $z_k$ ($k = 1, 2, \ldots, K$). Images were reconstructed using the mean time-of-flight of the time-resolved measurements. Numerical and phantom experiments demonstrated that the reconstructed area with changes in the absorption coefficient decreased as $p$ approached zero.

Prakash et al. reported the $\ell_p$-norm ($0 \leq p \leq 1$) regularization for CW domain measurement [113]. They implemented the majorize–minimization framework, which replaced the original nonconvex cost function with the $\ell_p$-norm ($0 < p \leq 1$) by a sequence of convex cost functions equal to the original cost function at a certain point and larger than the original cost function at the other points. Moreover, the regularization minimizing smooth $\ell_0$-norm was introduced. The smooth $\ell_0$-norm was approximated as $\|\delta\mu_a\|_0 = K - \sum_{k=1}^{K} \rho(\delta\mu_{ak})$ with $\rho(\delta\mu_{ak}) = \exp(-\delta\mu_{ak}^2/\sigma^2)$, which takes values of unity and zero when $\Delta\mu_{ak} > \sigma$ and $\Delta\mu_{ak} < \sigma$, respectively. Improvements in image quality were demonstrated in a numerical experiment with irregular geometry and a gelatin phantom experiment with optical properties mimicking typical breast tissue. Various applications of sparsity regularization have been proposed [114,115].

Although regularization minimizing the norm of change in the optical properties effectively reduces blurriness and can locate the localized cerebral blood flow change caused by neural activity and cancer tissues with angiogenesis in the early stage, it is difficult to reconstruct a certain volume of a relatively large tissue/organ with nearly uniform optical properties. In such cases, the total variation (TV) method may be useful. The TV regularization method minimizes the norm of the image derivative and reconstructs the image using a nearly piecewise constant with jump discontinuities [116]. The TV regularization term is represented as

$$R_{TV}(\mu) = \int |\nabla\mu(\mathbf{r})| d\mathbf{r}, \tag{21}$$

where $\mu$ is the reconstructed distribution of optical properties; $|\cdot|$ indicates the Euclidean ($L_2$) norm. In the actual calculation of the TV norm, a difference approximation of the gradient of $\mu$ is performed. By minimizing the gradient of $\mu$ (approximated with differences in $\mu$ on neighboring discretized nodes for numerical calculation using FEM), the image tends to be flat and has piecewise constant parts.

Paulsen and Jiang applied the TV regularization method to the FD-domain DOT to reconstruct $D$ and $\mu_a$ simultaneously [117]. Image reconstruction was attempted through numerical simulation with additive random noise and in the phantom experiment, where the ratios of the imaging target with a radius of 12.5 mm to the background were 2:1 and 10:1, respectively. Image reconstruction with TV regularization also reconstructed the area and values of $D$ and $\mu_a$ correctly, whereas image reconstruction without TV regularization was unable to correctly reconstruct the area and values.

Douiri et al. compared Tikhonov, TV, and Huber regularizations [118]. Although TV regularization reconstructs a volume with piecewise constant optical properties, it is difficult to reconstruct the changes in optical properties for a small volume included in a large volume. To overcome this difficulty and reconstruct small and large structures, the Hubert regularization term is formulated as follows:

$$R_{Hubert}(\mu) = \int \psi(\mu(\mathbf{r})) \, d\mathbf{r}, \tag{22}$$

$$\psi(\mu) = \begin{cases} |\nabla\mu|^2/2 & \text{if } |\nabla\mu| \leq \sigma \\ \sigma|\nabla\mu| - \sigma^2/2 & \text{otherwise} \end{cases}, \tag{23}$$

where $\sigma$ is the parameter for switching $\Psi$, which is automatically adjusted during each iteration. In the flat homogeneous part with a small gradient of the reconstructed image, $\Psi$ functioned similarly to the Tikhonov regularization to smooth the image while preserving the discontinuities with a large gradient. The numerical simulations demonstrated that the changes in $\mu_a$ and $\mu'_s$ of the small area in a large flat part was reconstructed more clearly by Hubert regularization than by TV regularization. The Hubert regularization method was extended to include prior information on the edges of regions in an object imaged using different imaging modalities [119].

### 4.2. Use of Structural Prior Information

Because DOT provides functional information, the interpretation of DOT images becomes more meaningful by superimposing DOT images onto morphological images obtained by other imaging modalities, such as X-ray CT, MRI, and ultrasound imaging. Moreover, information regarding the location where the optical properties can change improves the precision and efficiency of image reconstruction by reducing the number of unknown variables to be reconstructed.

Schweiger and Arridge proposed a two-stage image reconstruction method that employs prior information [120]. They assumed that the anatomical structure inside a measured object can be obtained using other imaging modalities. The averages of the optical properties in certain regions obtained by the segmentation of other images were reconstructed in the first stage. In the first stage, the optical properties of each region were assumed to be homogeneous. Subsequently, the details of the optical properties in each region were reconstructed in the second stage by employing the image obtained in the first stage as an initial estimate. It was demonstrated that the proposed two-stage method employing a segmented MR image of the brain provides a better image than image reconstruction with a homogeneous flat initial estimate.

Dehghani et al. incorporated prior information into the matrix $J$ [121]. The discretized positions (FEM node) in a segmented region were lumped by introducing a $K \times K$ matrix $U$ that had elements $U_{\xi,\eta} = 1$ when the $\xi$-th position was included in $\eta$-th segmented region, and was otherwise 0. The matrix $J$ in Equation (9) was replaced by the matrix $JU$. The ill-posed nature of the inverse problem was relieved by grouping the discretized

positions. The accuracy of the reconstructed values improved significantly in the 3D image reconstruction of small objects. The prior information that was introduced into $U$ to fix a homogeneous change in the optical properties in a segmented region is called the hard prior information.

Ntziachristos et al. employed a hard prior method to quantify hemoglobin concentration and oxygen saturation of breast lesions in 14 subjects [122]. They obtained prior structural information from the MR images. Image reconstructions of $\mu_a$ at wavelengths of 780 and 830 nm were carried out for the regions of the tumor and other tissues (background). The cancerous tumor had a hemoglobin concentration almost 10 times higher than that of the background tissues and the oxygen saturation of cancerous tissues was approximately 10% lower than that of the background tissues.

Boverman et al. investigated the influence of imperfect prior information, which are errors in the segmentation of MIR images, on two-step image reconstruction combined with background estimation with nonlinear optimization and linearized image reconstruction with a hard prior method [123]. By comparing the image reconstructed from the homogeneous initial guess with that from the initial guess obtained based on prior background information, it was demonstrated that image reconstruction with imperfect prior information can localize the abnormal region in the breast but causes a bias in the reconstructed image.

Di Sciacca et al. tried using hard prior information from ultrasound images in TD measurements with eight wavelengths ranging from 635 to 1060 nm [124]. Digital phantoms of the breast were used to visualize benign and malignant lesions. The digital phantoms were generated using software from the Virtual Imaging Clinical Trials for Regulatory Evaluation (VICTRE) Project. B-mode ultrasound images of the breast were simulated using $k$-wave software [125] by employing phantoms. The optical properties were determined using Gaussian random variables, which had different averages and variances between the benign and malignant lesions. For image reconstruction, segmentations were performed to separate the lesion and normal regions based on the B-mode image. Two-region image reconstruction was performed, and the reconstructed optical properties were used for classification, in which 75% of the lesions were classified correctly.

In contrast, the prior information incorporated into the regularization term was called soft prior information [25]. In this approach, prior information was incorporated into matrix $L$ in Equation (9). One of the methods for employing $L$ as prior information is to use the elements of $L_{k,k'} = 1$ when $k = k'$, $L_{k,k'} = 0$ when $k \neq k'$, and $L_{k,k'} = -1/K$ when the $k$-th and $k'$-th FEM nodes are in the same segmented region (Laplacian structure). Using this method, the difference in optical properties at positions in the same segmented region was minimized. Therefore, the optical properties tended to be homogeneous in each segmented region.

Yalavarthy et al. examined soft and hard prior methods [126,127] in numerical simulations with an MRI-based breast model and an experiment with a gelatin phantom. The Laplacian and Helmholtz structures were used as soft prior methods. In the Helmholts structure, which is a modified version of the Laplacian structure, $L_{k,k'} = -1/(K + h/l)$ when the $k$-th and $k'$-th FEM nodes are used in the same segmented region, where $h$ and $l$ are the distances between the nodes and the size of the imaging target (tumor), respectively. Compared with image reconstruction without prior information, the use of structural prior information dramatically improves the quantification and image quality of DOT.

Brooksby et al. used the soft prior method with MR to reconstruct the optical properties of the breast using FD measurements [128]. In numerical simulations, the soft prior method with a tissue layer structure recovered 74% of the true values. Additionally, a good initial estimate of the optical properties provided 99% recovery.

Li et al. proposed a method that uses prior information on the structure obtained using other imaging modalities in the regularization term [129]. Using the diagonal matrix $L$, which has a diagonal element of 1 corresponding to the voxel in the segmented tumor

region and 0 corresponding to the normal region, the cost function with the regularization terms was formulated with linearization in the forward process as

$$\|\delta \mathbf{M} - J\delta \boldsymbol{\mu}\|^2 + \gamma_1 \|(I - L)\delta \boldsymbol{\mu}\|^2 + \gamma_2 \|L\delta \boldsymbol{\mu}\|^2. \tag{24}$$

By taking a large value of the regularization parameter $\gamma_1$ and a small value of $\gamma_2$, $\delta\mu$ in the tumor region can become relatively large, although the artifacts, which are noises that appear in the reconstructed image, can be enhanced. In this study, the spectral prior method described in Section 4.3 was also employed.

The Bayesian framework incorporates prior structural information. Guven et al. proposed a hierarchical Bayesian approach [130]. Anatomical images, such as MRI and X-ray CT, were segmented into subimages composed of background tissues. In each subimage, we assumed that the absorption coefficient was Gaussian distributed with an unknown mean and variance, which are referred to as the hyperparameters. The FD measurements were then modeled with linearization employing the Rytov approximation and additive Gaussian noise. The hierarchical Bayesian approach estimated $\delta\mu_a$ and the hyperparameters simultaneously by maximizing the log-likelihood of the posterior probability density function.

In a manner different from the Bayesian framework, Panagiotou et al. treated DOT image reconstruction with probability statistics to incorporate prior structural information [131]. For the regularization term, they used the joint entropy and mutual information of the reconstructed and reference images obtained by different imaging modalities. The joint entropy was formulated as follows:

$$H(\delta\mu, x_{ref}) = -\iint p(\delta\mu, x_{ref}) \cdot \log\Big(p(\delta\mu, x_{ref})\Big) d\mu dx_{ref}, \tag{25}$$

where $p(\delta\mu, x_{ref})$ is the joint probability function of the image intensities $\delta\mu$ and $x_{ref}$ in the reconstructed and referenced images approximated by the intensity histograms. Entropy $H(x)$ is the expected amount of information by obtaining $x$, which is defined in information theory. Because $H(\delta\mu, x_{ref}) = H(\delta\mu \mid x_{ref}) + H(x_{ref})$, minimization of $H(\delta\mu, x_{ref})$ leads to a DOT image $\delta\mu$ providing a small amount of additional information $H(\delta\mu \mid x_{ref})$ after obtaining the reference image $x_{ref}$. This means that the DOT image behaved similarly to the reference image. In addition, the mutual information $MI(\delta\mu, x_{ref}) = H(\delta\mu) + H(x_{ref}) - H(\delta\mu \mid x_{ref})$, which is a measure of the similarity between $\delta\mu$ and $x_{ref}$, was maximized as the regularization term. In this case, $H(\delta\mu \mid x_{ref})$ was maximized, whereas the entropy $H(\delta\mu)$ indicating the randomness of $\delta\mu$ was minimized. It was demonstrated that $H(\delta\mu \mid x_{ref})$ is superior to $MI(\delta\mu, x_{ref})$ as the regularization term in numerical experiments because $MI(\delta\mu, x_{ref})$ constrained the solution more than $H(\delta\mu, x_{ref})$.

### 4.3. Use of Spectral Prior Information

While DOT can image the hemoglobin concentration and oxygen saturation of tissues by using the reconstructed $\mu_a$ and $\mu_s'$, DOT image reconstruction faces the cross-talk problem caused by nonuniqueness, which means that a reconstructed image with a combination of $\mu_a$ and $\mu_s'$ different from the true values is obtained from the measurements because the different combinations of $\mu_a$ and $\mu_s'$ cause the measurements to be identical or very close to each other.

Arridge and Lionheart [132] summarized the conditions for combinations of $(\mu_{a1}, D_1)$ and $(\mu_{a2}, D_2)$ that cause the same CW measurements, as follows:

$$\eta_0(\mu_{a1}, D_1) = \eta_0(\mu_{a2}, D_2) \text{ in } \Omega_0, \tag{26}$$

with

$$\eta_0(\mu_a, D) = \left[\left(\nabla^2 D^{1/2}\right)/D^{1/2}\right] + (\mu_a/D), \tag{27}$$

and

$$D_1 = D_2 \text{ in } \Omega_1, \tag{28}$$

where $\delta_0$ is the measured region surrounded by $\Omega_1$ including isotropic light sources.

It was reported that the cross-talk problem was reduced by the image reconstruction method, which directly reconstructed the concentrations of photon absorbers using multi-spectral measurements because the spectral prior information of photon absorbers could function as a constraint to maintain the consistency of the spectra in the reconstructed images [133,134]. In DOT image reconstruction employing prior spectral information, $\mu_a$ at the $o$ wavelengths, $\lambda_1, \lambda_2, \ldots \lambda_o$, was formulated with the concentrations of $q$ photon absorbers, $\mathbf{c(r)} = [c_1, c_2, \ldots c_q]^T$, as follows:

$$\begin{bmatrix} \mu_a(\lambda_1, \mathbf{r}) \\ \vdots \\ \mu_a(\lambda_o, \mathbf{r}) \end{bmatrix} = \begin{bmatrix} \varepsilon_1(\lambda_1) & \cdots & \varepsilon_q(\lambda_1) \\ \vdots & \cdots & \vdots \\ \varepsilon_1(\lambda_o) & \cdots & \varepsilon_q(\lambda_o) \end{bmatrix} \begin{bmatrix} c_1(\mathbf{r}) \\ \vdots \\ c_q(\mathbf{r}) \end{bmatrix}, \tag{29}$$

where $\varepsilon_1, \varepsilon_2, \ldots \varepsilon_q$, are the extinction coefficients of the photon absorbers. In contrast, $\mu_s'$ is defined as $\mu_s'(\lambda, \mathbf{r}) = a(\mathbf{r}) \cdot \lambda^{-b(\mathbf{r})}$, by assuming a simplified Mie-scattering coefficient, where $a$ and $b$ are the parameters depending on the size, refraction index, and concentration of the scatterers [133,135]. By substituting the equations relating $\mu_a$ and $\mu_s'$ to $a$, $b$, and $c$ into Equation (1) or (8) and constructing matrix $J$, image reconstructions with prior spectral information were performed.

Corlu et al. proposed an image reconstruction method for multispectral (four wavelengths) CW measurements using nonlinear gradient-based optimization [133]. In this study, $b$ was assumed constant, and $a$ and $\mathbf{c}$ were reconstructed. To reduce the cross-talk problem, the condition of increasing nonuniqueness in the multispectral-CW DOT was derived based on the conditions in Equations (26)–(28). $(a, \mathbf{c})$ and $(a + \delta a, \mathbf{c} + \delta \mathbf{c})$ yield the same measurements when the following equation holds.

$$\begin{bmatrix} \frac{\varepsilon_1(\lambda_1)}{\lambda_1^b} & \cdots & \frac{\varepsilon_q(\lambda_1)}{\lambda_1^b} \\ \vdots & \cdots & \vdots \\ \frac{\varepsilon_1(\lambda_o)}{\lambda_o^b} & \cdots & \frac{\varepsilon_q(\lambda_o)}{\lambda_o^b} \end{bmatrix} \left( \frac{\delta a}{a \cdot h(a + \delta a)} \begin{bmatrix} c_1 \\ \vdots \\ c_q \end{bmatrix} + \frac{1}{h(a + \delta a)} \begin{bmatrix} \delta c_1 \\ \vdots \\ \delta c_q \end{bmatrix} \right) = \begin{bmatrix} 1 \\ \vdots \\ 1 \end{bmatrix}, \tag{30}$$

where $\delta \mathbf{c} = [\delta c_1, \delta c_2, \ldots \delta c_q]^T$ and $h(a + \delta a) = \{1/3(a + \delta a)\} \cdot [\nabla^2 (1/3a)^{1/2}/(1/3a)^{1/2} - \nabla^2 \{1/3(a + da)\}^{1/2}/\{1/3(a + \delta a)\}^{1/2}]$. Equation (30) was rewritten using the vector matrix formula $A\mathbf{x} = \mathbf{1}$. The least-squares solution of the equation is $\mathbf{x}_0 = (A^T A)^{-1} A^T \mathbf{1}$ when $o > q$. The residual error norm, $E = \|A\mathbf{x}_0 - \mathbf{1}\|$, was used as the measure of the ability to distinguish $\mu_a$ from $\mu_s'$. When the residual error norm was close to zero, the measurement involved cross-talk problems in image reconstruction. In the numerical simulation, it was demonstrated that the cross-talk between $\mu_a$ and $\mu_s'$ was reduced by a combination of wavelengths that provided an $E$ that was as large as possible. Simultaneously, the concentrations of oxy- and deoxyhemoglobin were clearly separated by the combination of wavelengths with the small condition number of matrix $A$ of $\varepsilon$ in Equation (30). The condition number is the ratio between the maximum and minimum singular values, and indicates the degree of separation ability among $c_1, c_2, \ldots c_q$. This image reconstruction method was applied to the FD DOT image of oxy-/deoxyhemoglobin, water, and $\mu_s'$ in a real breast tissue at four wavelengths: 690, 750, 786, and 830 nm [136].

Li et al. used a spectral prior method for linearized image reconstruction [134]. The linear equation, $\delta \mathbf{M} = J\delta \mathbf{c}$, was solved for the changes in the concentrations of oxy- and deoxyhemoglobin, and $\delta \mathbf{c}$ was the regularized Moore–Penrose generalized solution, $\delta \mathbf{c} = J^T (JJ^T + \lambda I)^{-1} \delta \mathbf{M}$. To compare the image reconstructions of $\delta \mathbf{c}$ using a spectral prior (direct method) and calculation from the reconstructed $\delta \mu_a$ (indirect method), a numerical simulation and a blood phantom experiment were conducted. By measuring the contrast-

to-noise (CNR) ratio (peak value of deoxyhemoglobin divided by the mean standard deviation of the reconstructed values in voxels) and cross-talk (changes in oxyhemoglobin concentration divided by deoxyhemoglobin concentration), we verified that the indirect method could improve the contrast-to-noise ratio and reduce cross-talk.

Li et al. employed the spectral prior method with a formulation of $\mu_s'$ that was different from other methods [137]. Based on the Mie theory and the assumption that the size of the scattering particle is Gaussian distributed, the *u*-th scattering particle has a reduced scattering coefficient:

$$\mu_{s_u}'(\lambda) = \int_0^\infty \{3Q_{scat}(v, n, \lambda)[1 - g(v, n, \lambda)]/2v\} f(v) dv, \tag{31}$$

$$f(v) = \left(1/\sqrt{2\pi\beta^2}\right) \exp\left\{-(v - \alpha_u)^2/(2\beta^2)\right\}, \tag{32}$$

where $v$ is the particle size, $n$ is the refractive index, $\alpha_u$ is the average particle size, $b$ is the standard deviation, and $Q_{scat}$ is the scattering efficiency. Then, $\mu_s'$ is formulated as

$$\mu_s'(\lambda, \mathbf{r}) = \sum_{u=1}^U \varphi_u(\mathbf{r})\mu_{s_u}'(\lambda), \tag{33}$$

where $\varphi_u$ is the volume fraction of the *u*th scattering particle. Equation (33) can be written as a vector matrix formula, similar to Equation (29). $\varphi_u$ was reconstructed with $c$ simultaneously. This algorithm was examined in numerical simulations with various geometries including particles with sizes of 150, 1000, and 6000 nm in a phantom experiment using blood, intralipid, and agar powder. The algorithm was also tested on the clinical measurements of patients with breast cancer.

### 4.4. Other Important Topics: Regularization Parameter, Artifacts, Local Minima, and AI

In DOT image reconstruction, the selection of the regularization parameter $\gamma$ is important, as is the case in other inverse problems. The reconstructed image cannot sufficiently approximate the true optical properties when $\gamma$ is excessively large because the squared error term in Equation (2) is not sufficiently small. However, the measurement noise, which causes artifacts in the reconstructed image, disturbs the reconstructed image when $\gamma$ is extremely small. Selecting an appropriate $\gamma$ is one of the important aspects in image reconstruction.

The L-curve and generalized cross-validation (GCV) methods are often used to solve inverse problems. The L-curve method finds $\gamma$ that minimizes the squared error term and regularization term in good balance by plotting the log of the squared residual error term versus the log of the regularization term with various $\gamma$, which often draws an L-shaped curve [111,114,115,138]. $\gamma$ allows the corner of the L-shaped curve to efficiently minimize both terms.

The GCV method in the case of Equation (9) with $W = L = I$ selects $\gamma$ that minimizes the GCV function,

$$GCV(\gamma) = \|\delta\mathbf{M} - J\delta\boldsymbol{\mu}(\gamma)\|^2 / \left[\text{trace}\left\{I - J\left(J^T J + \gamma I\right)^{-1} J^T\right\}\right]^2, \tag{34}$$

where $\delta\boldsymbol{\mu}(\gamma)$ is the Tikhonov regularized image reconstructed using $\gamma$ [139,140]. The trace in Equation (34) is the sum of the diagonal elements in the matrix, which can be regarded as a measure of the degrees of freedom in the regularized image $\delta\boldsymbol{\mu}$ [116]. Various methods have been proposed to select $\gamma$.

Correa et al. examined 10 methods to select $g$ including heuristic, L-curve, GCV, unbiased predictive risk estimator, discrepancy principle, normalized cumulative periodogram (NCP), F-slope, quasi-optimality criterion, full width half maximum, contrast-to-noise ratio (CNR), and CNR·$\Psi^{-1}$ methods [141]. Linearized 3D image reconstruction was performed

using a tissue-mimicking phantom made of epoxy resin with the imaging target. The optimal method was selected based on the relative error in the reconstructed image, objectivity, and the requirement of no prior knowledge. Although it is difficult to find the corner of the L-shaped curve owing to its high ill-posedness, the L-curve method was selected as the optimal method under the conditions of the study.

Jagannath and Yalavarthy compared the minimal residual method (MRM) with GCV [139]. MRM updates $\delta\mu$ in a manner similar to that of the steepest gradient method to solve Equation (8) with $W = I$ using Tikhonov regularization with $L = I$. MRM changes $\gamma$ in each iteration to update $\delta\mu$. The updating rule at the $t$th iteration is

$$\delta\boldsymbol{\mu}_{t+1} = \delta\boldsymbol{\mu}_t - s_t \cdot \nabla f_t, \tag{35}$$

$$\nabla f_t = J^T(J\delta\boldsymbol{\mu}_t - \delta\mathbf{M}) - \gamma_t \cdot \delta\boldsymbol{\mu}_t, \tag{36}$$

where the step size of $s_t = \|\nabla f_t\|^2 / \left(\|J\nabla f_t\|^2 + \gamma_t\|\nabla f_t\|^2\right)$. $\gamma_t$, which minimizes the squared residual error, was searched using a simplex-type method. The image reconstructions based on the Rytov approximation with CW measurements in the numerical simulations and the gelatin phantom experiment demonstrated that the MRM provided a higher spatial resolution and peaks closer to the true values than the GCV method.

Prakash and Yalavarthy proposed the conjugate gradient-type least-square QR (LSQR) method to select $\gamma$ [142]. The LSQR method is typically used to solve linear equations efficiently and is applied to image reconstruction. The LSQR, MRM, L-curve, and GCV methods were compared using numerical simulations and gelatin phantom experiments. The LSQR and MRM methods improved the spatial resolution and quantitativeness of the reconstructed images.

Sun et al. examined methods to select $\gamma$ automatically, such as the L-curve, GCV methods, MRM, projection error method (PEM), and model function method in nonlinear image reconstruction using Tikhonov regularization from the logarithm of light intensities in CW measurements [143]. The PEM calculates $\gamma$ at the $t$th iteration, as follows:

$$\gamma_t = \max\left(J_t{}^T J_t\right) / \left[2 + \exp\left(-\|\mathbf{M} - \mathbf{F}_t\|^2\right)\right]. \tag{37}$$

The absolute bias error (ABE), which is the mean value of the error between the reconstructed and true values; CNR, which compares the values in the region of interest and background; and FWHM were evaluated through numerical simulations. PEM provided better CNR and ABE than the other methods, especially for the image reconstruction of a single target. MRM improved the spatial resolution more than the other methods.

Pogue et al. proposed $\gamma$ as a spatial variant [144]. Because of the diffusive nature of light propagation, the spatial resolution of the DOT image degrades as the distance from the target to the sources and detectors increases. A spatially constant $\gamma$ can excessively smooth the image of a deeper target. For image reconstruction of a circular object with Tikhonov regularization, the spatially variant $\gamma$ is formulated as follows:

$$\gamma\left(r_j^n\right) = \gamma_e \exp\left(r_j^n / R\right) + \gamma_c, \tag{38}$$

where $r_j^n$ is the radial position of the $j$th pixel and R is the radius of the image. $\gamma_e$ and $\gamma_c$ are empirically determined parameters. In the numerical simulation of the FD measurements with three targets, the spatially variant $\gamma$ reconstructs the target close to the center of the circular object more clearly than the spatially constant $\gamma$. The spatially variant $\gamma$ provided a better image than the spatially constant one by evaluating the peak value of the reconstructed target, the FWHM, and the variance in the homogeneous region. Phantom experiments were also conducted. The artifacts appearing in the high-sensitivity region close to the sources and detectors were suppressed using this regularization method.

Another way to suppress the artifacts and influence of measurement noise on the reconstructed image is to reduce the measurement noise. Okawa et al. proposed a noise reduction method for TD measurement [145]. The number of photons measured using the time-correlated single-photon counting method is a Poisson-distributed random variable with a mean equal to the noise-free photon count. Based on the probability density function of the DTOF measurements formulated using the Poisson distribution function, a noise-free DTOF that maximizes the posterior log-likelihood was estimated. Using the estimated noise-free DTOF, the artifacts and variations in the homogeneous region were reduced in the numerical and phantom experiments.

Errors in the measurement conditions cause a mismatch between the forward process and the actual measurement. Consequently, artifacts may appear in the image to compensate for this mismatch. Therefore, calibration of the sources and detectors is important for accurately measuring light intensities. Boas et al. demonstrated the simultaneous reconstruction of $\delta\mu$ and calibration factors [146]. In this study, the intensity of the $i$th light source at position $\mathbf{r}_{s,i}$ has a constant calibration factor $s_i$ and is formulated as $s_i \cdot q_0(\mathbf{r}_{s,i})$. Then, the measurement by the $j$th detector with calibration factor $d_j$ was expressed as $s_i \cdot d_j \cdot M_{i,j}$. The image reconstruction of the calibration factors and $\delta\mu$ was performed by linearization using the Rytov approximation. The matrix $J$ in Equation (7) can be rewritten to include the effects of $s_i$ and $d_j$: The matrix was formulated using Green's function for slab geometry. Reconstructions of $\delta\mu$ and the calibration factors were performed using Tikhonov regularization. $\delta\mu$ was scaled to be dimensionless for the simultaneous reconstruction. In the numerical experiment with the simulated measurements generated with a randomly chosen calibration factor, no artifacts appeared in the image reconstructed by the proposed method, whereas the images reconstructed without considering the calibration factor were contaminated with artifacts.

Oh et al. reported image reconstruction with a calibration factor (coupling coefficient) in FD measurement using a Bayesian framework updating $\mu_a$ and coupling coefficients sequentially [147]. The forward process used the finite difference method. A numerical simulation was performed for a cubic object. The phantom experiments employed a light-emitting diode and a photodiode as the source and detector, respectively. A cylindrical object in a culture flask filled with intralipids was reconstructed.

Schweiger et al. reported the nonlinear reconstruction of $\mu_a$ and $\mu_s'$ and the coupling coefficient to modify the errors in the amplitude and phase in FD measurements [148]. $\mu_a$, $\mu_s$, and the coupling coefficients were scaled by averaging the initial estimates. The FEM was used in the forward process. The damped Gauss–Newton method was used in the optimization process. The 2D numerical simulation and a cylindrical phantom experiment with two targets were performed. In the phantom experiment, hair was placed between the optical fibers and the phantom surface to simulate functional brain imaging.

Fukuzawa et al. applied a calibration coefficient to TD measurements [149]. The measured amplitude of the DTOF was influenced by changes in the contact condition of the optical fibers on the surface of the measured object, especially when the surface consisted of a soft material such as the skin of the scalp. The mGPST method using the Laplace-transformed DTOF was employed for image reconstruction. In addition to the numerical experiments, a phantom experiment employing a cylindrical polyacetal resin coated with soft silicone rubber demonstrated that the change in compression on the surface by the optical fibers caused an artifact at the surface in the reconstructed image. The calibration coefficient was applied to the in vivo measurement of the infant's head and reduced artifacts.

The estimation of the calibration coefficient prior to image reconstruction was proposed by Li and Jiang [150], employing a calibration matrix obtained from a homogeneous phantom, and by Tarvainen et al. [151] using a rotationally symmetric array of source–detector optical fibers in FD measurement.

In addition to the instability caused by noise, which is relieved by regularization, the reliability of a reconstructed image is influenced by its nonuniqueness. In particular,

using the gradient-based optimization method mentioned in Section 2.3, an initial guess of the optical properties may lead to a local minimum of the cost function, which causes the reconstructed image to not approximate the true image. Li et al. used a combination of a genetic algorithm (GA) and a gradient-based method in the optimization process to simulate DOT image reconstruction for prostate cancer diagnosis [152]. The GA randomly generated wide-ranging candidates for the reconstructed image, which were updated by mutation, crossover recombination, and selection [153]. The updating rule mimics the biological evolutionary process. The initial guess selected by the GA was used in the nonlinear gradient-based image reconstruction. The FEM geometry, including the prostate, intraprostatic tumor, and rectum, was constructed to solve the PDE based on endorectal MRI images.

Jiang et al. used simulated annealing (SA) to prevent the reconstructed image from being trapped by local minima [154]. In this method employing SA, which is a type of a Metropolis–Hastings Markov chain Monte Carlo method, $\delta\mu_a$ was expressed as a discrete value as $\delta\mu_a = \delta\mu_a{}^{\max} \{(S_i/M) + (1/2)\}$ by employing the spin variable $S_i = \pm 1, \pm 2, \ldots, \pm M/2$ at the $i$th position. The cost function $f(S_i)$ with linearization by Rytov approximation and a type of Tikhonov regularization was used in the Boltzmann distribution with temperature $T$, formulated as $p(S_i) \propto \exp(-f(S_i)/T)$, as the probability density function to sample the spin variables. $S_i$ randomly generated in accordance with $p(S_i)$ was adopted when the probability increased (i.e., $f$ decreased). Starting from a high $T$, which provided a highly random selection of $S_i$ and cooling by decreasing $T$, the spins resulting in the reconstructed image were narrowed from a wide range of candidates. The numerical simulations demonstrated that the SA avoided the local minimum arising in a single-spin case [155] and that the DOT image was reconstructed in the multiple-spin case.

AI, as represented by deep learning, can provide a different approach to reconstructing DOT images. Deep learning comprises an artificial neural network (ANN) with several hidden layers. Various ANN architectures for deep learning, such as convolutional neural networks (CNN), generative adversarial networks, and recurrent neural networks (RNN), have been employed. The neural network outputs the data processed by artificial neuron units, the connections of which are adjusted by a learning process using training data that are given input and output pairs. Deep learning and its relationship with the DOT have been detailed in the literature [16,17]. Yoo et al. proposed a deep-learning approach [15] that employed an ANN to solve the following Lippman–Schwinger equation for $\delta\mu$:

$$\delta\Phi(\mathbf{r}) = -\int_\Omega G_0(\mathbf{r}, \boldsymbol{\xi})\delta\mu(\boldsymbol{\xi})\Phi(\boldsymbol{\xi})d\boldsymbol{\xi}, \tag{39}$$

where $\delta\Phi = \Phi - \Phi_0$, and $\Phi_0$ and $G_0$ are the fluence rate and Green's function of the CW version of the PDE with the squared diffusive wave number of the background substituted for $\mu_a$. Equation (39) can be inverted using a technique related to Hankel matrix decomposition, which is achieved using a fully data-driven neural network with an encoder–decoder structure. The training data were generated by numerical simulations using NIRFAST. This approach output the image of $\delta\mu$ better than the $\ell_1$-sparse regularization method and the iterative updating method based on Rytov approximation in the numerical, phantom, and in vivo mouse experiments.

Mozumder et al. proposed an ANN output of the updated optical properties for the Gauss–Newton method used in a nonlinear optimization process [33]. Image reconstruction was formulated using the FD PDE. The Bayesian approach was employed to incorporate prior information on the distribution of the optical properties. In the process to update the optical properties, CNN was employed. The update of the optical properties is expressed as

$$\boldsymbol{\mu}_{t+1} = G_{\theta,t}(\boldsymbol{\mu}_t, \delta\boldsymbol{\mu}_t), \tag{40}$$

where $\boldsymbol{\mu}_t$ is the distribution of $\mu_a$ and $\mu_s{}'$ in the $t$th iteration, denoted by the subscript, $G_{\theta,t}$ is the updating function achieved by the CNN, and $\delta\boldsymbol{\mu}_t$ is the gradient of the cost function. $\boldsymbol{\mu}_t$ and $\delta\boldsymbol{\mu}_t$ are the inputs to the CNN. The selection of the step size, which must be

determined in the updating process of the nonlinear optimization, was reduced. Numerical experiments demonstrated that the proposed method worked better than the conventional Gauss–Newton method under modeling errors. The proposed method reduced crosstalk artifacts, whereas the Gauss–Newton method reconstructed images contaminated by artifacts.

Takamizu et al. proposed image reconstruction using the long short-term memory (LSTM) deep learning method, which is an extension of the RNN that uses gates to selectively retain and forget information [156]. The deep learning scheme is composed of two LSTM layers and a dense layer to reconstruct the positions with changes in $\mu_a$. The datasets generated using TD RTE were used to train the ANN. By comparing the DOT images reconstructed using the proposed method and the conventional nonlinear optimization method with DA, it was demonstrated that the proposed method reduced the blurriness of the region with a large $\mu_a$.

As shown in the above studies, the use of AI technology represented by deep learning can reduce computational costs in the forward and optimization processes using training data prior to practical image reconstruction. Forward calculation, particularly with RTE, which is repeatedly required in the nonlinear optimization process, requires massive computation, and becomes a bottleneck in the clinical application of DOT. AI can improve the quality of an image by confining the solution space of the inverse problem of DOT image reconstruction in accordance with appropriate prior information.

## 5. Discussion

The image-reconstruction algorithms reviewed in the previous sections were studied to address each technical problem. In actual clinical measurements, image reconstruction is required to consider these problems comprehensively. In particular, an improvement in the spatial resolution is essential for the diagnosis of cancers, which are important imaging targets of DOT. Studies on breast cancers using DOT and NIR spectroscopy indicated that malignant breast cancer had statistically different optical properties from normal tissues [8,10,12]. Malignant tumors can have a higher total hemoglobin concentration and a larger scattering coefficient than normal tissues and benign tumors. In addition to breast cancer, cancers of other organs involve abnormal vascularization. Microvessel density in tumors, which can be imaged noninvasively by diffuse optical techniques, can be an indicator of tumor grade and stage, and a prognostic indicator [157]. The progression and expansion of malignant tumors require nutrient supply and waste removal, which can be accomplished by microcirculation owing to angiogenesis and vasculogenic mimicry in tumors. Therefore, contrast with hemoglobin must be useful.

However, it is difficult to distinguish cancer from benign lesions using hemoglobin as a contrast agent because the hemoglobin concentration increases in both the carrier and benign lesions [12]. One of the important characteristics useful for cancer diagnosis is tumor heterogeneity. Although the hemoglobin concentration is an important biomarker, it changes temporally and spatially in tumors. The peripheral area and invasive edge of the tumor can have more blood flow and higher hemoglobin concentrations. However, blood flow in growing tumors can decrease due to decreasing vessel density, severe structural and functional abnormalities of the vessels, and the development of necrosis [158]. Heterogeneous hypoxic regions of tumors with low blood flow, which often exist in the core of tumors and resist radiotherapy, are of interest in various medical imaging studies [159]. By imaging the heterogeneity of tumors quantitatively and qualitatively using an improved image reconstruction algorithm, DOT may be applied as expected in cancer diagnoses.

Therefore, the low spatial resolution of DOT due to diffusive light propagation should be overcome by image reconstruction for clinical use, although it may not be easy for the current DOT to reconstruct the heterogeneity within a tumor a few centimeters in size. In image reconstruction, the mismatch between the forward calculation and the actual light propagation degrades image quality. Therefore, high-precision computation in the forward process is desirable. The use of RTE is a promising direction, especially

for imaging cancers and abnormalities such as thyroid cancer and rheumatic arthritis, which are near the illumination positions. The error of DA increases in regions with $\mu_a > 0.01$ mm$^{-1}$ and a distance smaller than 5 mm from the illuminating position, according to studies [74,97] comparing DA and MC simulations. DA can increase the error in the void and low scattering regions, such as the trachea in the neck and cerebral spinal fluid layer in the head, as discussed in the literature on RTE, $P_N/SP_N$ approximations, and the MC method. However, the use of the RTE with appropriate discretization is a time-consuming and computationally expensive process. Higher-order approximations using $P_N/SP_N$ methods [57–66] and hybrid methods [87] can also be useful.

In the application of DOT, including cancer imaging and functional brain imaging, the imaging targets associated with changes in hemoglobin concentration must be selected from a heterogeneous background consisting of multiple organs and different tissues. The effects of background heterogeneity cannot be ignored under realistic scenarios. Therefore, in the optimization process, methods used in the literature, such as [118], which can manage broad and local changes in the optical properties simultaneously, are needed. To obtain a reliable image, prior structural information from other imaging modalities such as MRI, X-ray CT, and ultrasound imaging is indispensable. Therefore, methods introduced in the literature [120,123,126,130], which incorporate prior structural information and recover the heterogeneity inside the segmented organs, should be employed. Because of the difficulty in selecting the regularization parameter, two-stage image reconstruction [120,123] incorporates hard prior information as an initial estimate. In the second stage, to reconstruct heterogeneities in the segmented regions, combination use of the sparsity regularization methods using $p$-norm ($0 \leq p \leq 1$) [104,106,109–111,113] can be useful to achieve the high-resolution image. The studies indicated that appropriate regularization improved not only the spatial resolution but also the quantification. Other biomarkers, such as water, lipids, collagen, and the scattering coefficient are useful for diagnosis, as shown in [136]. The spectral prior method improved the quantification performance of the DOT.

The combined use of DOT and other optical imaging methods is one approach for obtaining a higher spatial resolution. Some studies combining QPAT/FDOT and DOT have been reported [160–166]. QPAT and FDOT are related to the DOT. Photoacoustic tomography (PAT) achieved a higher spatial resolution than DOT by exploiting the low scattering characteristic of ultrasound excited by NIR light [167]. FDOT can enhance the contrast of malignant lesions using fluorescent molecules [47]. Both PAT and FDOT highlighted tissue abnormalities. However, the background optical properties, which are not observed in the PA and FDOT images, can be imaged by DOT. Moreover, the background optical properties are useful for quantifying PAT/FDOT images because the image intensity of PAT/FDOT depends on the fluence rate of the excitation light, which in turn depends on the optical properties of the heterogeneous background. The reconstruction of the optical properties of the PA sources [50] and the fluence compensation modification of the PA image [168–171] by QPAT will be improved using background optical properties quantified by DOT, which will play a very important role in guaranteeing quantification by PAT/FDOT. In this context, the aforementioned two-stage DOT image reconstruction method is useful. A quantitative comparison between abnormalities imaged by PAT/FDOT and normal tissues imaged by DOT may provide useful information for diagnosis.

## 6. Conclusions

DOT with NIR light, which can image the concentrations of chromophores such as hemoglobin deep inside the body, is expected to be a new imaging modality for cancer diagnosis and functional brain imaging. Unlike X-ray CT, DOT image reconstruction must consider diffusive light propagation, which reduces the spatial resolution and quantification of images. Studies on image reconstruction have attempted to resolve the technical problems of spatial resolution and quantification accuracy in DOT images of highly heterogeneous measured objects. Various image reconstruction methods were reviewed here.

In the forward process, the use of the RTE, $P_N$, $SP_N$, diffusion approximations, and MC simulations has been reported. Improvements in computational performance in recent years have enabled us to base the high-precision forward simulation of light propagation on predicting measurements for DOT image reconstruction. To reduce the mismatch between the forward simulation and the actual measurement, which degrades the quality of the reconstructed image, RTE and its higher-order approximation are useful. To balance the required precision in clinical applications with computational time and cost, an appropriate forward computation method should be chosen.

During the optimization process, various methods, including regularization techniques and the use of structural and spectral prior information, have been proposed to overcome the technical problems of low spatial resolution, noise, and artifacts. Sparsity regularization techniques have improved spatial resolution. The use of prior information on the structures inside the body obtained from other imaging modalities, such as X-ray CT and MRI, is useful for treating heterogeneous backgrounds. Spectral prior information improves the quantification of chromophore concentrations in multispectral DOT imaging. For reliable image reconstruction in cancer diagnoses with high heterogeneity, the use of prior information and sparsity regularization must be effective. By considering the purpose of DOT in clinical use and the specificity of the diffuse optical measurement system, an effective optimization method can be developed.

Although the authors have reviewed important studies, many of them could have been missed owing to the limitations of the authors' knowledge and time. As discussed in this review, various efforts have been made during the past two decades to improve image reconstruction. Moreover, technologies surrounding DOT, such as AI, high-performance computation, and PA imaging, have begun in recent years to change the circumstances. Novel solutions that apply emerging technologies to improve image reconstruction and DOT applications in clinics can be anticipated by leveraging numerous previously reported studies.

**Author Contributions:** S.O. and Y.H. reviewed the literature and edited the manuscript. All authors have read and agreed to the published version of the manuscript.

**Funding:** This research was supported by JSPS KAKENHI (grant number 21K04085).

**Institutional Review Board Statement:** Not applicable.

**Informed Consent Statement:** Not applicable.

**Data Availability Statement:** Data sharing is not applicable to this article.

**Conflicts of Interest:** The authors declare no conflict of interest.

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
