# Peer review of "A Review of Image Reconstruction Algorithms for Diffuse Optical Tomography"

_applsci, doi:10.3390/app13085016_

Round 1
Reviewer 1 Report
Comments and Suggestions for Authors
The paper is a rather bloated introduction than a full-fledged review.
I would like to see a theoretical justification for choosing algorithms and the classification used.
In conclusion, I would like to see more intelligible practical recommendations for selecting algorithms for different researches.
Author Response
Response to Reviewer 1
The authors would like to extend our appreciation for the reviewer’s time and effort on reviewing our manuscript. Thanks to the comments and suggestions, the manuscript has been improved significantly.
The point-by-point responses are listed as follows:
Point 1: The paper is a rather bloated introduction than a full-fledged review.
We intended to introduce the recent efforts on the DOT image reconstruction as much as possible. The following sentence is added from line 87 of the last paragraph in the end of the introduction section on page 2.
“The authors tried to cover as wide a range of topics and studies in this review as possible, although that may have made this review bloated rather than fully fledged.”
Point 2: I would like to see a theoretical justification for choosing algorithms and the classification used. In conclusion, I would like to see more intelligible practical recommendations for selecting algorithms for different researches.
We added the Discussion section from line 1084 on page 23, in which we discuss the theoretical justification and intelligible practical recommendations for selecting algorithms for DOT applications in clinic.
In the forward process, the mismatch between the forward calculation and the actual light propagation de-grades image quality. Therefore, the use of RTE is a promising direction, especially for imaging cancers and abnormalities such as thyroid cancer and rheumatic arthritis, which are near the illumination positions. The error of DA increases in regions with ma > 0.01 mm-1 and a distance smaller than 5 mm from the illuminating position, according to studies comparing DA and MC simulations.
In the optimization process, because of the heterogeneity of the background tissues, it becomes difficult to image the abnormality in the tissues with high spatial resolution and quantitative accuracy. Therefore, we recommend the two-stage algorithm to reconstruct the background with the hard prior obtained by the other imaging modalities in the first stage and localize the abnormalities in the second stage. In the second stage, it must be useful to use the sparsity regularization method minimizing Lp-norm of the reconstructed image to obtain high resolution image of the abnormality.
In the 2nd and 3rd paragraphs (from lines1172 to 1190) of Conclusions, the above discussion is summarized.
Finally, the authors thank the reviewer for the contribution to this manuscript again.

Reviewer 2 Report
Comments and Suggestions for Authors
In this manuscript, a variety of image reconstruction methods are reviewed. In the forward process, uses of the radiative transfer equation, PN, SPN, and diffusion approximations, and Monte Carlo simulation have been reported. Improvements of computational performances in recent years make it possible to bask in high precision of the forward simulation of light propagation predicting the measured data for Diffuse Optical Tomography (DOT) image reconstruction. For balancing the required precision in clinical application and computational time and cost, appropriate forward computation method should be chosen. In the optimization process, various methods including the regularization techniques and uses of structural and spectral prior information have been proposed to overcome the technical problems of low spatial resolution, noise, and artifact. By considering the purpose of DOT in clinical use and the specificity of the diffuse optical measurement system, the effective optimization method can be found. Although the authors reviewed important works, many of other important works have not been presented. Image reconstruction algorithm is a key technology to realize clinical uses of DOT and the other noninvasive optical imaging. By leveraging numerous recent researches in the literature, further advances in the image reconstruction algorithm can be expected as well as in the hardware and application developments for the Diffuse Optical Tomography.
In my point of view, this review manuscript is good enough to be considered in this journal. It was well written and presented. However, if they had discussed more researches in the study's direction it would be more reliable. Moreover, there are some English mistakes for example in line 1011 the ward '' To should be replaced by "For".
Author Response
Response to Reviewer 2
The authors would like to extend our appreciation for the reviewer’s time and effort on reviewing our manuscript. Thanks to the comments and suggestions, the manuscript has been improved significantly.
The point-by-point responses are listed as follows:
Point 1: In my point of view, this review manuscript is good enough to be considered in this journal. It was well written and presented. However, if they had discussed more researches in the study's direction it would be more reliable.
We add the section of the discussions from line 1084 on page 23 to line 1163 on page 25. We discussed the image reconstruction algorithms for the cancer imaging which is the important studies’ direction of DOT with newly added literatures [156-172].
One of the important characteristics useful for cancer diagnosis is tumor heterogeneity. By imaging the heterogeneity of tumors quantitatively and qualitatively using an improved image reconstruction algorithm, DOT may be applied as expected in cancer diagnoses. Therefore, the low spatial resolution of DOT due to diffusive light propagation should be overcome by image reconstruction for clinical use.
In the forward process, the mismatch between the forward calculation and the actual light propagation degrades image quality. Therefore, the use of RTE is a promising direction, especially for imaging cancers and abnormalities such as thyroid cancer and rheumatic arthritis, which are near the illumination positions. The error of DA increases in regions with a > 0.01 mm-1 and a distance smaller than 5 mm from the illuminating position, according to studies comparing DA and MC simulations. However, the use of the RTE with appropriate discretization is a time-consuming and computationally expensive process. Higher-order approximations using PN/SPN methods [57-66] and hybrid methods [87] can also be useful.
In the optimization process, because of the heterogeneity of the background tissues, it becomes difficult to image the abnormality in the tissues with high spatial resolution and quantitative accuracy. Therefore, we recommend the two-stage algorithm [120,123] to reconstruct the background with the hard prior obtained by the other imaging modalities in the first stage and localize the abnormalities in the second stage. In the second stage, it must be useful to use the sparsity regularization method minimizing Lp-norm [104,106,109,110,111,113] of the reconstructed image to obtain high resolution image of the abnormality.
The combined use of DOT and other optical imaging methods, such as quantitative photoacoustic tomography (QPAT) and fluorescence diffuse optical tomography (FDOT), is one approach of DOT study to obtain higher spatial resolution [161-167]. The background optical properties are useful for quantifying PAT/FDOT images because the image intensity of PAT/FDOT depends on the fluence rate of the excitation light, which in turn depends on the optical properties of the heterogeneous back-ground. The reconstruction of the optical properties of the PA sources [50] and the fluence compensation modification of the PA image [169-172] by QPAT will be improved using background optical properties quantified by DOT, which will play a very important role in guaranteeing quantification by PAT/FDOT.
In the 2nd and 3rd paragraphs (from lines1172 to 1190) of Conclusions, the above discussion is summarized.
Additionally, use of artificial intelligence technology, which is the new direction of DOT image reconstruction, is described at the end of section 4.4 from line 1031 to 1083 with newly added literature [156].
Point 2: Moreover, there are some English mistakes for example in line 1011 the ward '' To should be replaced by "For".
The revised manuscript was proofread by the professional native English speakers of Editage, Cactus Communications, Inc. (manuscript reference number is ZJKPV_3Z) ( https://www.editage.jp/ ).
Finally, the authors would like to show our appreciation to the reviewer’s contribution to this manuscript again.

Reviewer 3 Report
Comments and Suggestions for Authors
Author Response
Response to Reviewer 3
The authors would like to extend our appreciation for the reviewer’s time and effort on reviewing our manuscript. Thanks to the comments and suggestions, the manuscript has been improved significantly.
The point-by-point responses are listed as follows:
Point 1: I strongly encourage the authors to perform an extensive editing of the english, maybe with the help of some native english speaker. …. Finally, the english of this paper is very confusing, with several typos and repetitions, and several sentences should be rewritten.
The authors thank the reviewer for the kind instructions on English.
The revised manuscript was proofread by the professional native English speakers of Editage, Cactus Communications, Inc. (manuscript reference number is ZJKPV_3Z). ( https://www.editage.jp/ ).
Point 2: The introduction is very well written but it lacks of important information like who may be interested in this review, to whom it is addressed, why it is important to publish another review and what are the present limitations on the field to be overcome and what is being done in this direction in the field. Similarly, also the conclusions should report again briefly these information, as well as many more details about a comparison between the various methods reported; which is best for what? Why a reader should chose one method instead of another one? Even if deductible from the various session, a summary about it is crucial for a review.
We added the descriptions about the background and purpose to write this review article in 4th paragraph from line 58 to lone 81 on page 2 as follows,
“Studies on image reconstruction have attempted to resolve the technical problems of spatial resolution and quantification accuracy in DOT images of highly heterogeneous measured objects. Image reconstruction has been improved by the recent studies mentioned in this review, although these efforts have not been reflected in clinical applications. To overcome this low spatial resolution, sparsity regularization techniques related to compressed sensing technology have been introduced. The use of prior in-formation about the structure inside the body obtained from other imaging modalities, such as X-ray CT and magnetic resonance imaging (MRI), has been proposed to improve spatial resolution. Spectral prior information improves the quantification of chromophore concentrations in multispectral DOT imaging. Moreover, changes in the circumstances surrounding DOT in the past two decades will affect image reconstruction algorithms. Recent progress in computational technology, including artificial intelligence (AI) and high-performance supercomputers, may have altered image reconstruction of DOT. Image reconstruction schemes employing AI with deep learning in diffuse optical imaging [15-19] and the simulation of light propagation using super-computers [20] have also been attempted in recent years. Progress in diffuse optics and related imaging technologies, including photoacoustic (PA) imaging, which allows high-resolution imaging of blood vessels deep inside the body [21,22], may promote reconsideration of the role of DOT and its image reconstruction. In such a changing situation, it is worth reviewing what has been achieved in DOT image reconstruction for research in diffuse optical imaging and related fields. This review will provide useful information to select algorithms for clinical applications of DOT and will assist re-searchers working in emerging DOT-related research fields, including AI, high-performance computation, and different optical imaging technologies, to expand their research into DOT.”
The above changes in the introduction are included in the conclusion section and the abstract.
Additionally, the discussion section is added in the revised manuscript from line 1084 on page 23 to line 1163 on page 25. In the section, we discuss what can be required in the image reconstruction for cancer diagnoses to attempt answering the question in the reviewer’s comment, “which is best for what? Why a reader should chose one method instead of another one?” The discussion is summarized as follows,
One of the important characteristics useful for cancer diagnosis is tumor heterogeneity. By imaging the heterogeneity of tumors quantitatively and qualitatively using an improved image reconstruction algorithm, DOT may be applied as expected in cancer diagnoses. Therefore, the low spatial resolution of DOT due to diffusive light propagation should be overcome by image reconstruction for clinical use.
In the forward process, the mismatch between the forward calculation and the actual light propagation degrades image quality. Therefore, the use of RTE is a promising direction, especially for imaging cancers and abnormalities such as thyroid cancer and rheumatic arthritis, which are near the illumination positions. The error of DA increases in regions with ma > 0.01 mm-1 and a distance smaller than 5 mm from the illuminating position, according to studies comparing DA and MC simulations. However, the use of the RTE with appropriate discretization is a time-consuming and computationally expensive process. Higher-order approximations using PN/SPN methods [57-66] and hybrid methods [87] can also be useful.
In the optimization process, because of the heterogeneity of the background tissues, it becomes difficult to image the abnormality in the tissues with high spatial resolution and quantitative accuracy. Therefore, we recommend the two-stage algorithm [120,123] to reconstruct the background with the hard prior obtained by the other imaging modalities in the first stage and localize the abnormalities in the second stage. In the second stage, it must be useful to use the sparsity regularization method minimizing Lp-norm [104,106,109,110,111,113] of the reconstructed image to obtain high resolution image of the abnormality.
The combined use of DOT and other optical imaging methods, such as quantitative photoacoustic tomography (QPAT) and fluorescence diffuse optical tomography (FDOT), is one approach of DOT study to obtain higher spatial resolution [161-167]. The background optical properties are useful for quantifying PAT/FDOT images because the image intensity of PAT/FDOT depends on the fluence rate of the excitation light, which in turn depends on the optical properties of the heterogeneous back-ground. The reconstruction of the optical properties of the PA sources [50] and the fluence compensation modification of the PA image [169-172] by QPAT will be improved using background optical properties quantified by DOT, which will play a very important role in guaranteeing quantification by PAT/FDOT.
In the 2nd and 3rd paragraphs (from lines1172 to 1190) of Conclusions, the above discussion is summarized.
Finally, the authors express our appreciation to the reviewer’s contribution to this manuscript again.

Round 2
Reviewer 3 Report
Comments and Suggestions for Authors
The authors replied to all my comments